# Quantitative control of noise in mammalian gene expression by dynamic histone regulation

**Deng Tan[1,2†], Rui Chen[1†], Yuejian Mo[1†], Shu Gu[1], Jiao Ma[1], Wei Xu[3], Xibin Lu[4], Huiyu He[1], Fan Jiang[5], Weimin Fan[1], Yili Wang[4], Xi Chen[3]\*, Wei Huang[1]\***

[1]School of Life Sciences, Southern University of Science and Technology, Shenzhen, China; [2]Department of Chemistry, The Hong Kong University of Science and Technology, Kowloon, Hong Kong, China; [3]Shenzhen Key Laboratory of Gene Regulation and Systems Biology, School of Life Sciences, Southern University of Science and Technology, Shenzhen, China; [4]Core Research Facilities, Southern University of Science and Technology, Shenzhen, China; [5]Department of Biomedical Engineering, Southern University of Science and Technology, Shenzhen, China

**Abstract** Fluctuation ('noise') in gene expression is critical for mammalian cellular processes. Numerous mechanisms contribute to its origins, yet the mechanisms behind large fluctuations that are induced by single transcriptional activators remain elusive. Here, we probed putative mechanisms by studying the dynamic regulation of transcriptional activator binding, histone regulator inhibitors, chromatin accessibility, and levels of mRNAs and proteins in single cells. Using a light-induced expression system, we showed that the transcriptional activator could form an interplay with dual functional co-activator/histone acetyltransferases CBP/p300. This interplay resulted in substantial heterogeneity in H3K27ac, chromatin accessibility, and transcription. Simultaneous attenuation of CBP/p300 and HDAC4/5 reduced heterogeneity in the expression of endogenous genes, suggesting that this mechanism is universal. We further found that the noise was reduced by pulse-wide modulation of transcriptional activator binding possibly as a result of alternating the epigenetic states. Our findings suggest a mechanism for the modulation of noise in synthetic and endogenous gene expression systems.

**\*For correspondence:**
chenx9@sustech.edu.cn (XC);
huangw@sustech.edu.cn (WH)

[†]These authors contributed equally to this work

**Competing interests:** The authors declare that no competing interests exist.

## Introduction

Isogenic cells in a homogenous environment can exhibit significant variations in gene expression. A single cell also shows similar fluctuations in gene expression over time. This phenomenon, often called gene expression noise, was initially demonstrated by computational modeling of stochastic biochemical reactions with finite biomolecules (*McAdams and Arkin, 1997*). *Elowitz et al., 2002* first designed experiments to investigate and characterize gene expression noise. Many general mechanisms that contribute to gene expression noise in mammalian cells have been identified, including partition at cell division, transcriptional bursting, epigenetic modifications, and 3D chromosome structure (*Huh and Paulsson, 2011*; *Nicolas et al., 2018*; *Rodriguez et al., 2019*; *Singer et al., 2014*; *Suter et al., 2011*). These studies often involved gene expression systems that are regulated by multiple transcriptional activators and complex interactions. Substantial noise is also observed in synthetic inducible gene expression systems involving just one transcriptional activator, often causing such systems to exhibit more noise than endogenous systems in mammalian cells (*Kyba et al., 2002*). However, the mechanistic origin remains to be identified experimentally. Various biological processes, such as cell fate control in proliferation, differentiation, and cell death, rely on precise control of gene expression (*Balázsi et al., 2011*). There are many examples in which

cells and animals benefit from either suppressing or elevating noise (*Chang et al., 2008*; *Hansen et al., 2018*; *Li et al., 2017*; *Sosnik et al., 2016*). Synthetic inducible gene expression systems, including Tet-Off (*Gossen and Bujard, 1992*), Tet-On (*Gossen et al., 1995*), chemical systems (*Khalil et al., 2012*), light-induced systems (*Wang et al., 2012*), and CRISPR-derived systems (*Nihongaki et al., 2017*; *Shao et al., 2018*) have been developed to perform perturbations of biological processes in cells and animals. With many toolboxes already developed in synthetic biology, we can now seek a better understanding of the complex biological processes that underlie fluctuations in gene expression by studying the quantitative and dynamic modulation of critical gene products in mammals. Synthetic inducible gene expression also enables mechanistic studies of the essential functions of transcriptional regulation by allowing dynamic and quantitative modulations, which are not possible in endogenous gene regulation systems.

An inducible expression system usually consists of an engineered transcriptional activator and an engineered promoter that includes a TATA box and multiple binding sites for the transcriptional activator. The transcriptional activator is usually constructed to include a DNA binding domain and the activation domain of an endogenous transcriptional activator, such as the p65 subunit (p65AD) of nuclear factor kappa-B (NF κ-B) or VP16 (VP16AD) from herpes simplex virus type I (*Gossen and Bujard, 1992*). Upon binding to the promoter, the transcriptional activator recruits coactivators, the mediator complex, and the transcriptional preinitiation complex (PIC) to initiate transcription. Such simple gene expression systems are often associated with large noise, although the mechanisms behind this noise are unclear. This limits the potential applications of inducible expression systems in situations that require the regulation of gene expression heterogeneity, such as studies of cell fate control. Several engineering approaches have been reported to suppress noise in doxycycline- (dox-) and light-induced expression systems in mammalian cells. Two engineered circuits were constructed with a dox-induced tetR-based negative feedback loop (*Nevozhay et al., 2013*) or a light-induced tetR-LOV negative feedback loop (*Guinn and Balázsi, 2019*). The potential applications of those circuits are limited because they have fixed noise levels and use the cell-type-sensitive CMV promoter.

In this study, we sought to use a LightOn inducible gene expression system to identify potential mechanisms behind the generation of large amounts of noise by a single transcriptional activator (*Wang et al., 2012*), and to develop strategies to control noise and mean expression independently. We generated stably transfected human ovarian cancer (HeLa) and mouse embryonal carcinoma (F9) cell clones with this LightOn expression system, using p65AD and dynamic and quantitative light inductions. We used flow cytometry to discover that pulse-width modulation (PWM) of the illumination with a period of 400 min or longer reduces gene expression noise. We identified histone acetylation as being key to the large fluctuations (noise) in amplitude modulation (AM) illumination.

We hypothesize that the light-activated transcriptional activator binds to the promoter in the open chromatin region in a stochastic manner and recruits the CBP/p300 coactivator. CBP/p300 not only facilitates the recruitment of PIC and initiates transcription but also acetylates histones in the vicinity, keeping the chromatin in an active/open state. This interplay could generate bimodality in histone acetylation and transcriptional activity under intermediate amplitude modulation (AM) illumination. We propose that illumination with PWM reduces noise by alternating the cell between 'high' and 'low' states with smaller heterogeneity. We used single-cell data to demonstrate the stability of these 'high' and 'low' states. We utilized A485, a specific inhibitor of CBP/p300 HAT activity (*Lasko et al., 2017*), a p65AD mutant that impairs p65AD-CBP/p300 interaction (*Mukherjee et al., 2013*), H3K27ac ChIP-seq analysis, single-cell transposase-accessible chromatin using sequencing (scATAC-seq) analysis (*Chen et al., 2018*), and live-cell single mRNA imaging (*Tutucci et al., 2018*) to validate the hypothesis. Furthermore, we combined the CBP/p300 inhibitor (A485) and the HDAC4 inhibitor (LMK-235) to disrupt the interplay between epigenetic regulation and transcriptional regulation involving CBP/p300 in mouse embryonic stem (mES), HeLa, and F9 cells, and observed reduced heterogeneity of endogenous gene expression for genes with high noise. These observations suggest that a noise modulation mechanism involving CBP/p300 could occur widely.

## Results

### Constant induction of a mammalian light-inducible system result in bimodality and large noise in gene expression

To facilitate studying the dynamic induction of gene expression, we stably integrated the LightOn expression system (*Wang et al., 2012*; *Figure 1A*) into HeLa cells using the PiggyBac transposon (*Lu and Huang, 2014*). The LightOn system consists of a synthetic transcriptional activator GAVPO and the mRuby3 (*Bajar et al., 2016*) reporter driven by a synthetic 5xUAS promoter consisting of five GAL4 binding elements and a TATA box (plasmid B1 in *Figure 1—figure supplement 1*). To assess the contribution of variation in GAVPO expression to the mRuby noise, the expression of GAVPO is driven by a noise reduction circuit of pCMV-tetO2-tetR-GFP-T2A-GAVPO (*Nevozhay et al., 2013*), where GFP provides an indirect measurement of GAVPO expression in individual cells (plasmid A1 in *Figure 1—figure supplement 1*). This approach also facilitates the regulation of mean GAVPO expression. The engineered HeLa cell clone is referred to as HeLa-AB1. In the following experiments, a doxycycline concentration of 1 µg/mL was chosen to ensure the cells were responsive to moderate light intensities. We measured expression of the fluorescent proteins by flow cytometry. At a constant light illumination (AM, illustrated in *Figure 1B*) with intensities between 0 and 100 µW/cm$^2$, the average level of mRuby expression in the population increased with increasing light intensity above 1 µW/cm$^2$ (blue line in *Figure 1C*). The gene expression noise was quantified by the coefficients of variation (CV). The cell population exhibited a bell-shaped noise profile across both the range of light intensities (blue line in *Figure 1D*) and the range of mean mRuby expression levels (blue line in *Figure 1E*). The highest noise level (CV ~2.5) was induced at intermediate light intensities (~10 µW/cm$^2$) (*Figure 1D*). The single-cell expression distributions showed a bimodal pattern, and an increasing number of cells transited from the 'low' to the 'high' state when the light intensity increased from 10 to 45 µW/cm$^2$ (solid lines in *Figure 1F*). A similar phenomenon was observed in the dox-inducible (Tet-On) system (solid line in *Figure 2—figure supplement 1B*).

Benzinger and Khammash proposed that in *Saccharomyces cerevisiae*, noise in light-induced gene expression is caused by noise transmission from transcription factor concentration to gene expression (*Benzinger and Khammash, 2018*). This noise can be modeled as a combination of a deterministic single-valued propagation function and an added phenomenological noise contribution, described as *Equation 1* in the Materials and methods section. The detailed calculations are described in the Materials and methods section and in Appendix 2. For AM induction at 25 µW/cm$^2$ (*Figure 1G*), the GFP signal exhibited a unimodal distribution with a CV of 0.47, shown in the lower panel of *Figure 1H*. Assuming GAVPO exhibits noise similar to that seen for GFP, a hypothetical sigmoid GAVPO-mRuby response curve with high cooperativity (black dashed line in the main panel in *Figure 1H*) could lead to high variability and a bimodal distribution in mRuby (shown as the black dashed line in the right panel in *Figure 1H*). To assess the contributions of GAVPO noise to mRuby distribution dispersion, we split the flow cytometry data (*Figure 1G*) into 20 bins based on the GFP signal. We then plotted a dose–response curve of the mean mRuby versus mean GFP-GAVPO expression for these 20 subpopulations (red circles) and fit these data to an empirical function (red dashed line in the main plot in *Figure 1H*). Using this shallow curve as the deterministic propagation function, Benzinger and Khammash's hypothesis would predict a unimodal distribution of mRuby with a relatively low CV value of 0.9, shown as the red dashed line in the right panel of *Figure 1H*. Therefore, the shape of the bimodal distribution and the high noise level (CV = 1.73) of actual mRuby expression in HeLa-AB1 cells (*Figure 1—figure supplement 2B*) cannot be explained by Benzinger and Khammash's hypothesis for *Saccharomyces cerevisiae*, at least if the assumption of similar noise for GFP and GAVPO in our system is correct. We further looked at the mRuby distributions of the 20 subpopulations, assessing both the experimental and the predicted data. The CVs for experimentally measured mRuby expressions exhibited a bell-shaped function and ranged between 1.0 and 1.7 (*Figure 1—figure supplement 2A*). The distributions for subpopulations transitioned from unimodal to bimodal and back to unimodal as mean GFP increased (*Figure 1—figure supplement 2C*). On the other hand, Benzinger and Khammash's model predictions with two single-valued propagation functions can only exhibit unimodal distributions (*Figure 1—figure supplement 2D–I*), which cannot explain our observations. This analysis suggested that, given a narrowly defined GFP-GAVPO level, the 5xUAS promoter could exhibit distinct 'high' and 'low' transcription states.

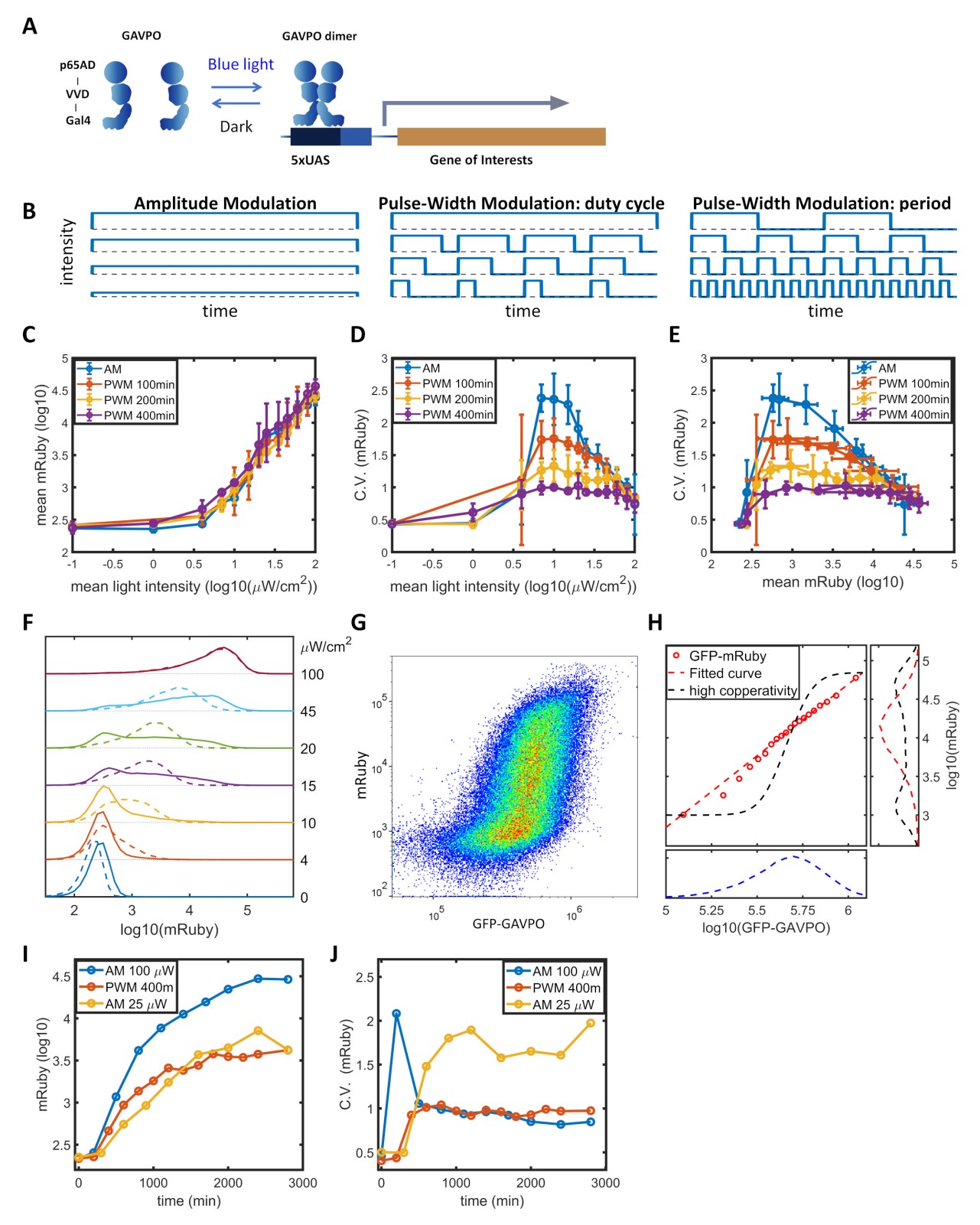

**Figure 1.** Gene expression and noise of the LightOn-mRuby expression system in HeLa-AB1 cells during light induction with amplitude modulation (AM) or pulse-width modulation (PWM). (A) Schematic representation of the LightOn expression system. (B) Illustration of AM and PWM light modulations. (C) The response curves of mean mRuby expression relative to mean light intensity for AM (blue) and PWM with a period of 100 min (red), 200 min (yellow), and 400 min (purple). (D–E) The correlation curves of the CV of mRuby expression versus mean light intensity (D) and mean mRuby

*Figure 1 continued on next page*

*Figure 1 continued*

expression (E) for AM (blue) and PWM with a period of 100 min (red), 200 min (yellow), and 400 min (purple). The error bars represent standard deviations from 2–4 independent experiments. (F) Analysis of AM-induced mRuby expression distribution (solid lines) and PWM-induced mRuby expression distribution (dashed lines, with a period of 400 min) for various mean light intensities, as marked on the right axis. (G) Density plot of mRuby against GFP-GAVPO expression under AM with 25 µW/cm². (H) In the large panel, a computed GAVPO-GFP versus mRuby response curve is shown with the cells from (G) divided into 20 equal populations in ascending order of GFP intensity; mean mRuby is plotted against mean GFP-GAVPO (both as common logarithms) (red open circles) and fitted to a Hill function (red dashed line) with a Hill coefficient of 1.87. A hypothetical highly cooperative curve is illustrated by the black dashed line. The lower panel shows the distribution of GFP-GAVPO expression. The predicted mRuby distributions based on the experimentally determined and hypothetical response curves are shown as the red and black dashed lines, respectively, in the right panel. All measurements were performed 48 hr after light induction. (I–J) The mean and CV of mRuby protein at different time points for AM with intensity of 100 µW/cm² (blue), for PWM with a maximum intensity of 100 µW/cm², a period of 400 min, and an on-fraction of 0.25 (red), and for AM with intensity of 25 µW/cm² (yellow).

The online version of this article includes the following source data and figure supplement(s) for figure 1:

**Source data 1.** Source data for *Figure 1*.
**Figure supplement 1.** Schematic of plasmids for the LightOn expression circuits.
**Figure supplement 2.** Comparison between experimental mRuby distributions and predictions of single-valued GAVPO-mRuby functions.
**Figure supplement 2—source data 1.** Source data for *Figure 1—figure supplement 2*.
**Figure supplement 3.** PWM modulation of distribution dispersion in HeLa-AB1 cells.
**Figure supplement 3—source data 1.** Source data for *Figure 1—figure supplement 3*.
**Figure supplement 4.** Noise modulation in HeLa-AB2 and F9-AB2 cells.
**Figure supplement 4—source data 1.** Source data for *Figure 1—figure supplement 4*.
**Figure supplement 5.** Noise modulation in HeLa-A1B1B3, HeLa-A4B5, and HeLa-A4B6 cells.
**Figure supplement 5—source data 1.** Source data for *Figure 1—figure supplement 5*.

## Modulation of gene expression noise with periodic induction

'Low' and 'high' noise levels might be related to epigenetic states at the 5xUAS promoter. The concept of epigenetic bistability was theorized by Sneppen et al. (*Dodd et al., 2007*; *Sneppen and Ringrose, 2019*). Still, experimental studies have mainly been concerned with long-term epigenetic memory (*Hathaway et al., 2012*; *Singer et al., 2014*) instead of fast gene expression fluctuations. *Bintu et al., 2016* demonstrated that the kinetics of different histone modifiers vary from hours to days in CHO cells. We hypothesized that by modulating the binding of the transcriptional activator periodically, with time scales similar to those of histone modification dynamics, the transcription dynamics might be 'resonated' with histone modification dynamics. In dynamic system theory, this type of resonance might reduce stochasticity (*Collins et al., 1995*; *Ding et al., 1994*). To explore this possibility, we use a pulse-width modulation (PWM, illustrated in *Figure 1B*) regiment with light switching off and on (between 0 and 100 µW/cm²), and the mean light intensity controlled by the fraction of 'on' time (duty cycle) with a fixed period. The period of 400 min results in minimal changes in mean gene expression (purple line in *Figure 1C*), but produces a significant reduction in gene expression noise across the intermediate mean light intensities and intermediate mean gene expression levels (purple lines in *Figure 1D–E*). For the intermediate mean light intensities, distribution spreading is reduced by 'eliminating' the 'low' and 'high' states, resulting in the generation of an intermediate peak, which increases with mean light intensity (dashed lines in *Figure 1F*). When the period of PWM is shortened, the noise levels (*Figure 1D-E*) and distributions gradually revert towards the AM scenario (*Figure 1—figure supplement 3A–F*). Therefore, we were able to modulate gene expression level and noise independently by altering the mean light intensity and the period of PWM within the range sandwiched between the blue and purple lines in *Figure 1D-E*. A preliminary lookup table is computed by fitting expression level and distribution spreading (quantified as the ratio of mRuby intensities at 90th and 10th percentiles) to mean light intensity and period of PWM (*Figure 1—figure supplement 3H,I*).

To confirm the discovery, we stably transfected HeLa and F9 mouse embryonal carcinoma cells with a LightOn-GFP circuit (plasmids A2, B2 in *Figure 1—figure supplement 1*) to obtain HeLa-AB2 and F9-AB2 clones. We observed similar phenomena of high noise with AM and reduced noise with PWM with HeLa-AB2 (*Figure 1—figure supplement 4A–C*) and F9-AB2 (*Figure 1—figure supplement 4D–F*) cells, suggesting that the modulation of LightOn gene expression noise by PWM could be common in mammalian cells. Mean light intensity and period of PWM do not act as orthogonal

parameters to control gene expression mean and noise. In F9 cells especially, expression level is often lower with PWM than with AM of the same mean intensity. To maintain the mean expression while adjusting noise, we needed to simultaneously adjust both mean light intensity and period of PWM (*Figure 1—figure supplement 3H*).

We further generated multiple clones of HeLa-A1B1B3 cells from Hela-AB1, each with similar GAVPO expression but different copy numbers of 5xUAS-mCardinal insertion sites (ranging in number from 3 to 11); these clones exhibited similar distribution dispersions with AM light (*Figure 1—figure supplement 5A B*). This observation suggested that bimodality is a robust phenomenon. To validate whether similar phenomena occur with chemical-induced p65 transactivator binding, we replaced the VP16 domain of rtTA on the classic tetON system with p65AD. We found that this system exhibited phenomena similar to those in LightOn systems (*Figure 1—figure supplement 5D*). We further reduced the tetO number from seven to one. However, this decreased the dynamic range of expression by ~200 to ~six fold, to levels too low to allow a clear definition of the existence of bimodality (*Figure 1—figure supplement 5C*). As singe-cell analysis of histone modifications is able to provide greater sensitivity than protein analysis, further developments in single-cell ChIP-seq assays could help to resolve this technical problem.

To characterize the dynamic noise reductions with PWM, we illuminated the HeLa-AB1 cell with AM at 100 µW/cm$^2$ or 25 µW/cm$^2$, or with PWM with a period of 400 min, and performed flow cytometry analysis over time. As shown in *Figure 1I*, the mean expression level of mRuby started to increase at 2–300 min and reached a plateau at around 2000 min for PWM and 2800 min for both AM intensities. The delayed onset of mRuby fluorescence is partially due to its maturation time of approximately 150 min. The noise levels remain relatively constant from 1000 min onwards (*Figure 1J*). Interestingly, for AM with 100 µW/cm$^2$, the noise level spiked initially before settling to a low level. These observations suggest that PWM with a long period is needed to maintain a low level of noise.

## Histone acetylation and transcription-epigenetic bimodality

Previous studies of histone modification kinetics in CHO cells suggested that histone deacetylase 4 (HDAC4) could be involved in our observed reduction in gene expression noise , as its rate constants are closer to the 400 min of PWM than those of other histone modifying enzyme (*Bintu et al., 2016*). To identify histone modifications that contributed to the noise in gene expression levels, we tested the effects of LMK-235, a selective inhibitor for HDAC4/5 (*Marek et al., 2013*). We also tested Vorinostat, a broad-spectrum inhibitor of HDACs (*Finnin et al., 1999*), and PF-06726304, a selective inhibitor of a Polycomb Repressive Complex 2 component and of H3K27me3 (*Kung et al., 2016*). We added these inhibitors to the HeLa-AB1 cell and performed light inductions for two days with AM at 25 µW/cm$^2$ before flow cytometry analysis. Only LMK-235 significantly increased the expression of mRuby (*Figure 2A–B*). To compare levels of mRuby expression at the same GAVPO levels, we calculated the population average mRuby versus GFP-GAVPO response curves for cells treated with these inhibitors (*Figure 2C*). The mRuby versus GFP-GAVPO curve for LMK-235 treated cells (red) is significantly steeper than that of the control cells (blue). Furthermore, this curve also demonstrated that LMK-235 explicitly increases the mRuby expression of cells with the same GAVPO level. When comparing the effects on AM and PWM with a 400 min period, we found that LMK-235 increased mRuby expression but had only a minor impact on noise reduction (*Figure 2—figure supplement 2A–F*). LMK-235 also increased gene expression levels for F9-AB2 cells and HeLa-Tet-on cells (*Figure 2—figure supplement 1A–B*).

The p65AD of the transcriptional activator GAVPO is reported to recruit CBP/p300. The dual-functional proteins CBP/p300 serve both as coactivators to recruit components of the mediator and transcriptional preinitiation complex (PIC) and as a histone acetyltransferase (HAT) (*Black et al., 2006*; *Gerritsen et al., 1997*; *Ogryzko et al., 1996*). A recent CBP/p300 acetylome study revealed their rapid acetylation kinetics at many loci in the mouse genome (*Weinert et al., 2018*). Putting this information together, we propose a mechanistic model for transcription-histone acetylation coupling of the light-inducible expression system (*Figure 2D*) in which: (a) chromatin stochastically switches between silent and active states; (b) light induces GAVPO dimerization and enable its binding to UAS elements of the 5xUAS promoter in the active chromatin; (c) bound GAVPO recruits CBP/p300; (d) CBP/p300 acetylates nearby histones, and recruits PIC and mediator complex to facilitate transcription initiation; and (e) a locus with high histone acetylation would tend to stay in the active state

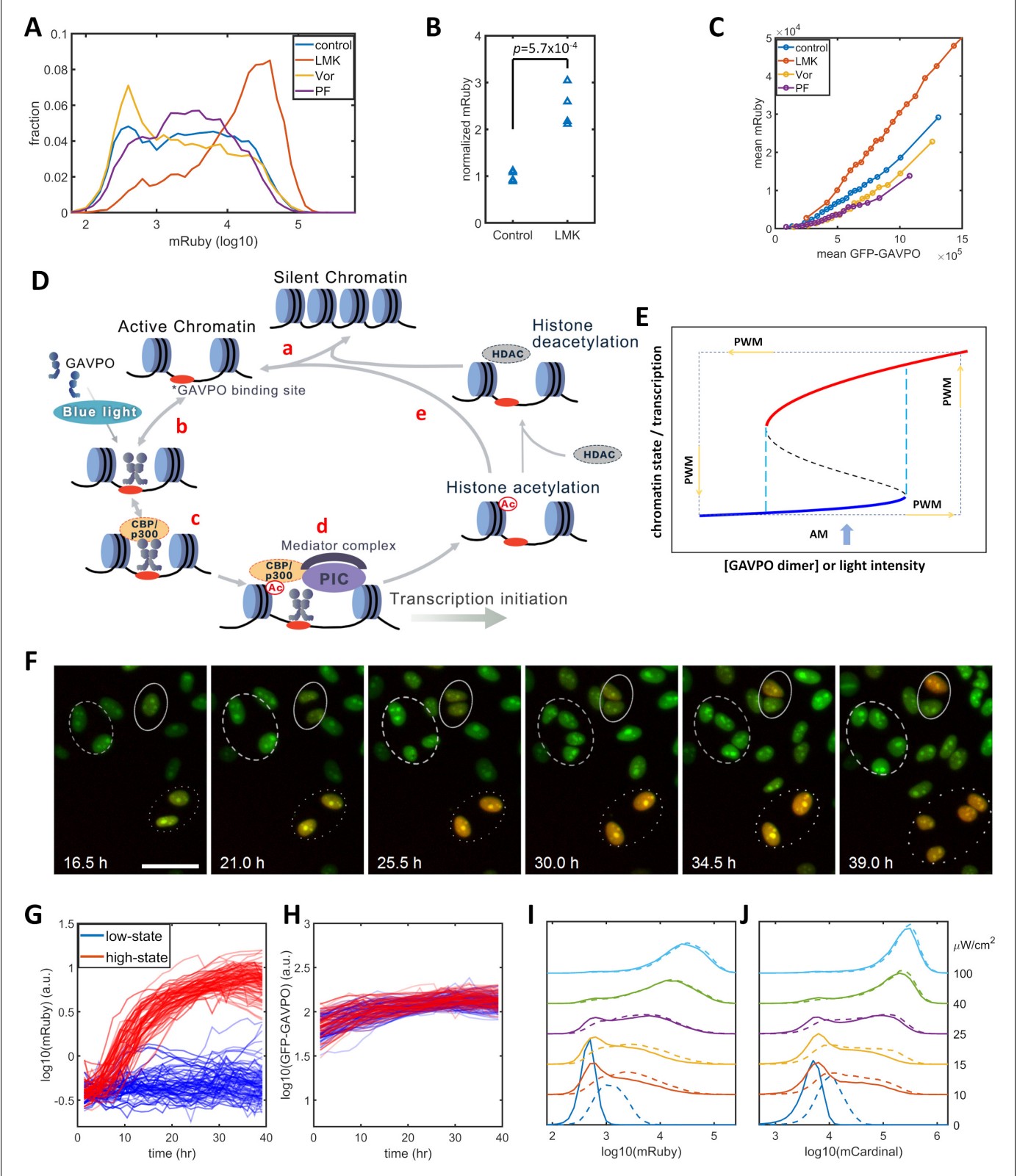

**Figure 2.** Model of light-induced gene expression and epigenetic bistability. (**A**) mRuby expression distribution of HeLa-AB1 cells under AM with 25 µW/cm[2] and treated with inhibitors of various epigenetic regulators. LMK, LMK-235 (a selective HDAC4/5 inhibitor); Vor, Vorinostat (SAHA) (a broad-spectrum inhibitor of HDACs); PF, PF-06726304 (a selective inhibitor for the EZH2 component of the Polycomb Repressive Complex 2 (PRC2)). The concentration of the three inhibitors was 0.333 µM. (**B**) Unpaired *t*-test for normalized mRuby expression comparing control and LMK-235-treated HeLa-

*Figure 2 continued on next page*

*Figure 2 continued*

AB1 cells. Data represent four independent experiments. (C) Response curves of population average mRuby expression versus population mean GFP-GAVPO expression after treatment with inhibitors of various epigenetic regulators. (D) Proposed mechanism underlying transcription and epigenetic regulation events involved in the expression of the LightOn system. 'HDAC' represents HDAC4/5 and possibly other HDACs; PIC, transcriptional preinitiation complex; Ac, histone acetylation. (E) Schematic view of the induction of bistability by a positive feedback loop, illustrating the induction of high noise levels by AM and the reduction of noise by PWM. The black dashed line represents the boundary that separates the high and low states, also known as the unstable steady state. The cyan dashed lines represent the thresholds between the low monostable state (blue) and bistable states (blue and red) and between the bistable states (blue and red) and the high monostable state (red). The blue arrow indicates an AM of intermediate intensity located at the bistability region. The PWM cycle (represented by the dashed rectangle) alternates cells between high and low monostable states. (F) Time-lapse images of single-cell mRuby expression dynamics for HeLa-A1B1 cells with 25 $\mu W/cm^2$ AM light induction. The green and red colors represent the intensities of nuclear-localized GFP-GAVPO and mRuby, respectively. Single nuclei were segmented and tracked with customized scripts and used to quantify single-cell expression of GFP-GAVPO and mRuby. The scale bar represents 50 $\mu m$. (G–H) Tracking and quantification of the single-cell dynamics of mRuby (G) or GFP-GAVPO (H) expression in HeLa-A1B1 cells under 25 $\mu W/cm^2$ AM light induction. Red and blue lines represent 100 cells in 'high' or 'low' states, respectively, after 24 hr. (I–J) Plots of mRuby (I) and mCardinal (J) expression distribution for HeLa-A1B1B4 cells with AM of increasing light intensity. Solid and dashed lines represent cells without or with prior 8 hr of 100 $\mu W/cm^2$ light inductions, respectively. Each sample contains 10,000–50,000 cells.

The online version of this article includes the following video, source data, and figure supplement(s) for figure 2:

**Source data 1.** Source data for *Figure 2*.
**Figure supplement 1.** Inhibition of HDAC4/5 in F9-AB2 and HeLa-Tet-On cells.
**Figure supplement 1—source data 1.** Source data for *Figure 2—figure supplement 1*.
**Figure supplement 2.** HDAC4/5 inhibition in HeLa-AB1 cells.
**Figure supplement 2—source data 1.** Source data for *Figure 2—figure supplement 2*.
**Figure supplement 3.** Normalization of single-cell expression dynamics.
**Figure supplement 3—source data 1.** Source data for *Figure 2—figure supplement 3*.
**Figure 2—video 1.** Single-cell mRuby-expression dynamics.
https://elifesciences.org/articles/65654#fig2video1

and have an increased chance of binding of GAVPO dimer. In summary, the DNA-GAVPO-HAT complex increases the probability of chromatin accessibility and GAVPO binding to DNA by increasing histone acetylation.

This hypothetical positive feedback loop could generate bistability, probably with intermediate GAVPO dimer concentration (or light intensity) (between the dashed cyan lines in *Figure 2E*). HDACs deacetylate histones and contribute to setting the boundaries between bistability and monostability. For some cells, the local promoter-GAVPO dimer interaction is sufficiently high as to initiate the positive feedback loop and to elevate local histone acetylation and induce high gene expression (red line). Other cells fail to start the positive feedback loop, which leads to low histone acetylation and transcription (blue line). Under this condition, the isogenic cells in a homogenous environment would exhibit large noise due to the history-dependent occupation of each state and stochastic switches between the two states (*Isaacs et al., 2003*). This could explain the wide distributions of mRuby expression with narrow GFP-GAVPO inputs (*Figure 1—figure supplement 2C*). The mechanism for PWM-induced noise reduction could be that the cell is switching between a high unimodal expression state (red) at 100 $\mu W/cm^2$ and a low unimodal expression state (blue) in the dark. In each state, the noise is low, and cells pass through the bimodal region rather quickly. For PWM with a shorter period, a cell doesn't have enough time to settle at either of two unimodal states, which leads to high noise.

We assessed the existence of high and low mRuby states by tracking the single-cell mRuby expression dynamics of HeLa-AB1 cells in 25 $\mu W/cm^2$ blue light induction using live-cell fluorescent microscopy. As both GFP and mRuby were tagged with three repeats of the SV40 nuclear localization signal, the GFP-GAVPO channel is used to segment and track cell nuclei over time. Single-cell expressions of GFP-GVPO and mRuby were integrated over the segmented nucleus. A representative time-lapse image series of the overlayed GFP and mRuby signals is shown in *Figure 2F*. The dashed and dotted circles in *Figure 2F* indicate that two cells remained in a low mRuby state through cell divisions, whereas two other cells had an elevated mRuby state which was maintained through cell division. The solid circle indicates that two offspring cells of one cell had different mRuby states after cell division. The mRuby expression states were designated as low (<0.1) or high (>0.6) after around 24 hr. The single-cell mRuby trajectories for 100 cells each from low (blue) and

high states (red), normalized by cell-cycle stage, are shown in *Figure 2G*. The normalization process is illustrated in *Figure 2—figure supplement 3*. Most of the cells at low mRuby expression state at 24 hr remain low through 40 hr, with the mRuby levels of a few cells being elevated toward the high state. Most of the cells at high mRuby state at 24 hr were elevated before 12 hr and remained in the high state through 40 hr. A small fraction of these cells show a drop in mRuby expression, moving them towards the low state. The GFP-GAVPO expression levels were within a factor of two and indistinguishable between the low-state and high-state cells (*Figure 2H*). These data indicate that the cells mostly maintain their mRuby expression states, with only a small fraction of them stochastically switching mRuby expression states. These single-cell dynamics are consistent with the bistability hypothesis, which assumes a low probability of random transitions. To validate bistability, longer live-cell imaging is required to observe sufficient occurrences of transition between the two states, partially because of the long half-life of the mRuby protein. At present, it is a challenge to maintain longer cell tracking without interruptions for periodic replacements of the culture medium.

To further assess the possibility of bistability, we examined its main characteristic, history-dependence. A cell in a bistable regime remains in the high or low state in a deterministic system, which is dependent only on the previous state of that cell. Stochastic fluctuations induce random transitions between the two states. When compared to 'naïve' cells that have not been exposed to blue light before the experiment, 'primed' cells that were exposed to 100 µW/cm$^2$ blue light immediately before the experiment would exhibit tilted distribution toward 'high' states under intermediate AM light induction. However, this priming would produce mRuby expression that partially remains after 48 hr. We constructed another plasmid B3 (*Figure 1—figure supplement 1*) containing destabilized mCardinal (mCardinal-PEST, shorter half-life) (*Li et al., 1998*) under the control of the 5xUAS promoter to increase its temporal response, and stably transfected this plasmid into HeLa-AB1 cell to generate HeLa-A1B1B4 clones. When compared to 'naïve' cells, the cells primed with 8 hr of 100 µW/cm$^2$ blue light exhibited visible but minor distribution shifts toward the high state after 48 hr of further AM blue light inductions at 10 to 25 µW/cm$^2$ before mRuby (*Figure 2I*) and mCardinal (*Figure 2J*) measurements. The relatively short time scale of CBP/p300 H3K27 acetylation increases stochastic transitions between the postulated 'low' and 'high' states and prevents observation of a more significant shift toward the high state as a result of priming.

## Disruption of epigenetic bimodality by inhibiting HATs and dynamic alternating chromatin accessibility

We propose that a high level of noise in gene expression is the consequence of epigenetic bimodality induced by the interplay among promoter-bound GAVPO, HATs, and histone acetylation. Consequently, a reduction in the occurrence of bimodality could be realized by weakening CBP/p300 HAT activities and breaking the postulated positive feedback. We validated this prediction by utilizing A-485, a highly selective and potent inhibitor of the HAT activities of CBP/p300 (*Lasko et al., 2017*; *Weinert et al., 2018*), to disrupt the dynamic regulation of these proteins without changing their transcriptional initiation function. When A-485 was added to HeLa-AB1 cells under AM light induction for 48 hr, the CV values for mRuby expression were significantly reduced, especially at intermediate light intensities and mean mRuby expression levels (*Figure 3B–C*). It seems that A-485 eliminates the 'low' and 'high' expression states and generates an intermediate state that has a narrower dispersion of mRuby expression (*Figure 3E*). This experiment also indirectly confirms that CBP/p300 are the prominent coactivators involved in transcription and histone modifications. The mean levels of mRuby expression were also reduced (*Figure 3A*), possible because HDACs now tilt the balance towards lower histone acetylation, leading to lower chromatin accessibility and less transcription. To validate this explanation, we added both A-485 and LMK-235 to cells under AM light induction for 48 hr. As shown in *Figure 3A-C, and F*, the addition of the HDAC4/5 inhibitor increased the expression level of these cells back to those seen in the light induction control, while the reduction in noise was retained. Hence, simultaneous inhibition of CBP/p300 HAT activity and HDAC4/5 could be used to reduce noise without changing mean gene expression. However, their effect on either synthetic or endogenous gene expression has never been examined.

To exclude the global effects of A485 and LMK-235, we generated a GAVPO mutant in which the direct binding of the p65 activation domain to CBP/p300 was greatly reduced by double point mutations corresponding to L449A/F473A in the TA2 section of the activation domain of mouse p65 (*Mukherjee et al., 2013*). We then constructed a plasmid A3 (PB5-CAG-GAVPO-mutant-PB3), and

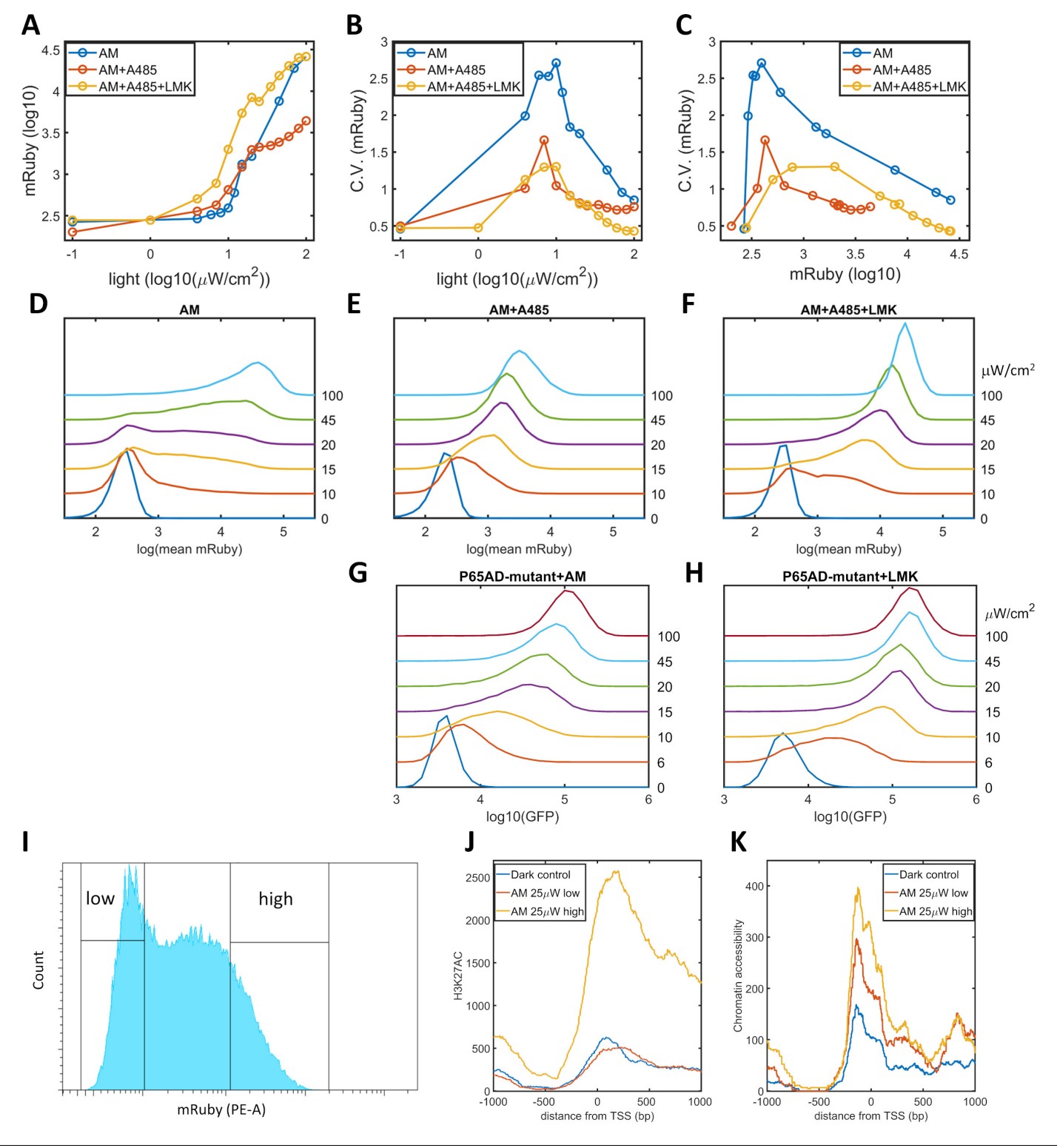

**Figure 3.** Validation of the epigenetic bistability of HeLa-AB1 cells in CBP/p300 inhibitor, H3K27ac, and ATAC-seq analyses. (**A–C**) Dose-response curve of mean mRuby expression versus light intensity (**A**), CV of mRuby expression versus light intensity (**B**) or versus mean mRuby expression(**C**). The experimental conditions include AM modulation alone (blue), AM with 1 μM A-485 (red), and AM with 1 μM A-485 and 0.333 μM LMK-235 (yellow). (**D– F**) mRuby expression distribution under increasing light intensities for AM alone (**D**), AM with 1 μM A-485 (**E**) and AM with 1 μM A-485 and 0.333 μM LMK-235 (**F**). The light intensities are specified to the right of each line. (**G–H**) GFP fluorescence distribution of HeLa-A3B2 cells at increasing light intensities for AM alone (**G**) and AM with 0.333 μM LMK-235 (**H**). (**I**) HeLa-AB1 cells treated with one day of AM light of 25 μW/cm²intensity were sorted

*Figure 3 continued on next page*

*Figure 3 continued*

into low- and high-mRuby-expression populations for further ChIP-seq and ATAC-seq analysis. (J–K) H3K27ac ChIP-seq (J) and ATAC-seq (K) analyses for the inserted 5xUAS promoter, aligned at the transcription start site (TSS), for the dark control (blue), a low-mRuby-expression population induced by AM 25μW/cm$^2$ (red), and a high-mRuby-expression population induced by AM 25μW/cm$^2$ (yellow). The dark control was measured in cells kept in the dark. Each sample contains 10,000–50,000 cells.

The online version of this article includes the following source data and figure supplement(s) for figure 3:

**Source data 1.** Source data for *Figure 3*.
**Figure supplement 1.** Insertion sites for LightOn cassettes.
**Figure supplement 2.** Emulation of combinations of 'high' states at multiple sites.
**Figure supplement 2—source data 1.** Source data for *Figure 3—figure supplement 2*.

generated a HeLa-A3B2 cell line. With an intermediate level of AM light induction for 48 hr, the disruption of the direct interaction between GAVPO and CBP/p300 also eliminated the 'low' and 'high' expression states. Instead, this treatment resulted in an intermediate state with narrow GFP expression dispersion (*Figure 3G*). Further inhibition of HDAC4/5 using LMK-235 increases GFP expression and its sensitivity to light intensity (*Figure 3H*). These two experiments suggested that p65-CBP/ p300 interaction and CBP/p300 activity in the proximity of the 5xUAS promoter are necessary for transcriptional bimodality. An additional mutation (F542A) in the TA1 section of the p65 activation domain disrupts the interaction of this domain with the Kix domain of CBP (*Lecoq et al., 2017*). However, this mutation also disrupts interactions with other critical transcriptional factor complexes, such as TFIIH, which might explain our inability to isolate HeLa cells with the p65AD mutants L449A/ F473A/F542A.

To assess the existence of epigenetic bimodality directly, we exposed HeLa-AB1 cells to AM light of 25 μW/cm$^2$ for one day to induce large expression dispersion. We then sorted the mRuby-low and mRuby-high populations (*Figure 3I*). We used the two cell populations and dark control cells to perform ChIP-seq assays for H3K27ac and mapped the reads to the whole genome sequence assembly of this HeLa-AB1 clone containing LightOn expression cassettes at nine loci (*Figure 3—figure supplement 1A–B*). The reads from these loci were combined using the TSS as the reference (*Figure 3J* and *Figure 3—figure supplement 1A*). We observe a H3K27ac peak spanning from the 5xUAS promoter (−200 bp) to the N-terminus of the mRuby gene. The signals for the mRuby-low and the dark control populations are essentially the same, whereas the signal for the mRuby-high population is about four times higher. These signals demonstrate that the mRuby-high population from intermediate light induction is in a more epigenetically active state than the mRuby-low and dark control populations. We also performed an ATAC-seq assay on the same cell populations. As shown in *Figure 3K*, chromatin accessibility is higher in close vicinity to the TSS (approximately from −200 to +200 bp). The hierarchy of chromatin accessibility is mRuby-high, mRuby-low, and dark control, with less separation between the high and low populations than H3K27ac and a clear difference between mRuby-low and dark control. Hence H3K27ac is correlated with the LightOn gene expression state, consistent with a critical role of CBP/p300.

To observe the effect of PWM induction on chromatin accessibility, we performed a single-cell ATAC-seq analysis of the dynamic process of PWM-induced noise reduction. Specifically, we set up a PWM regimen with 100 μW/cm$^2$ maximum light, 600 min period, and 25% duty cycle (150 min 'on' time). We collected cells at six timepoints corresponding to different light cycle stages (as illustrated in *Figure 4A*), and sorted 384 tagged nuclei of each population for sequencing and analysis. The mean number of reads spanning −200 to +200 bp exhibited a gradual increase over each light induction cycle, (*Figure 4B*). The single-cell reads for the six populations are plotted as heatmap images *in Figure 4C*. As the opening and closing of chromatin is a dynamic process, we only observed approximately 30% of cells with reads at most. As expected, the fraction of cells with open chromatin was lowest for the dark control. Each cycle of 'on' light (150, 750, 1350 min) leads to more cells with detected reads. At the end of each 'on-off' cycle (600, 1200 min), the fraction of cells with detected reads did not go all the way down to the level of the dark control. To obtain quantitative measurements, we took the mean reads over 400 bp for each cell and calculated the means and CVs for all populations, which are plotted against time in *Figure 4D–E*. The mean number of reads started at around 0.7, increased after each 'on' phase, and decreased to a lesser extent after each 'off' phase. The CV for the mean reads was initially at a high level of 3.8 (high noise of chromatin

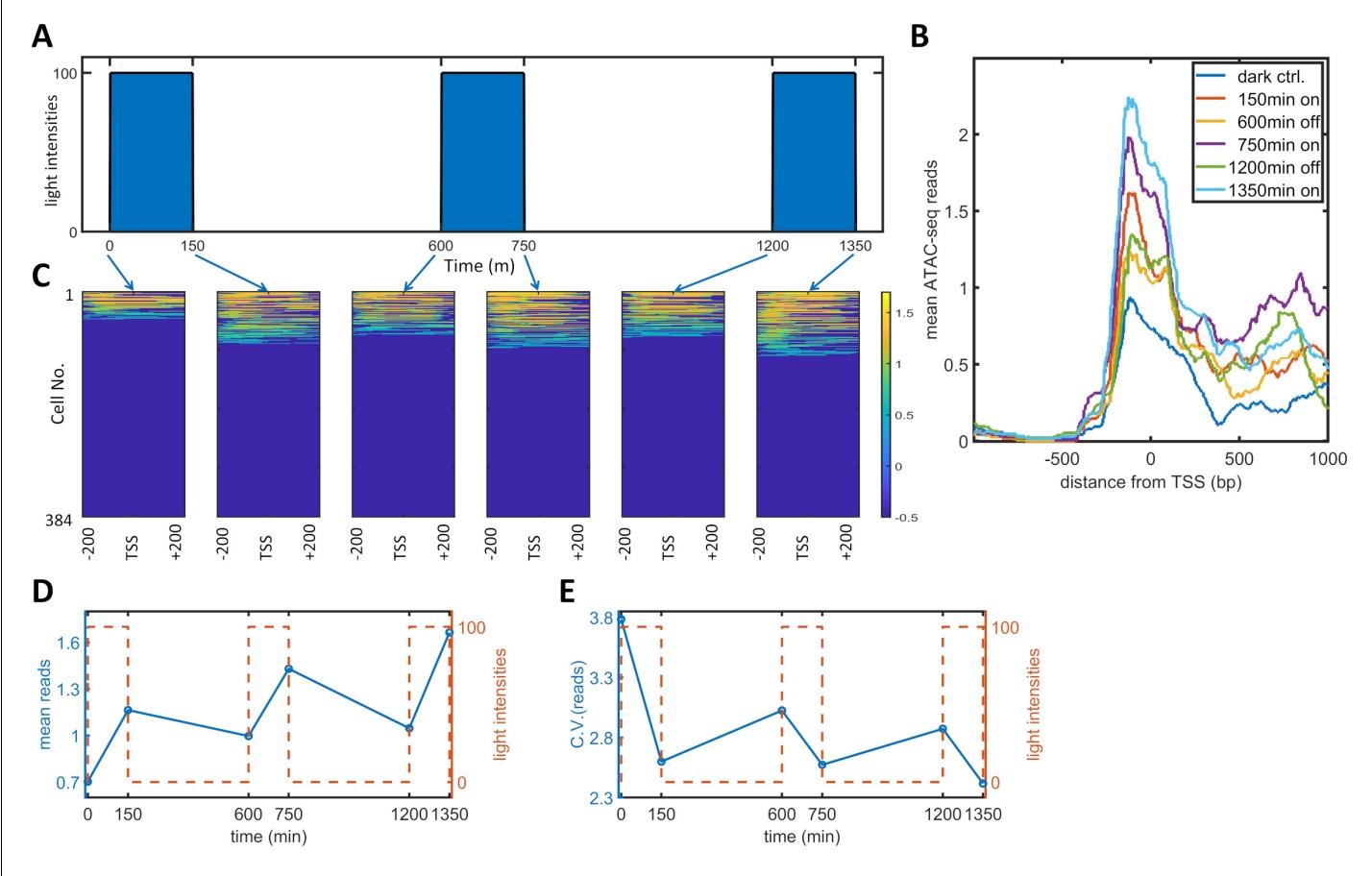

**Figure 4.** Single-cell ATAC-seq analysis of dynamic noise reductions for HeLa-AB1 cells with PWM. (**A**) A dynamic PWM induction program was used to generate cells for single-cell ATAC-seq. As indicated with blue arrows, six cell populations were collected: first at dark control (0 min), and then after one light-on cycle (150 min), after one light-on-off cycle (600 min), after one light-on-off cycle and one light-on cycle (750 min), after two light-on-off cycles (1200 min), and after two light-on-off cycles and one light-on cycle (1350 min). (**B**) The average number of ATAC-seq reads from a single cell plotted against distance from the transcription start site (TSS) of the 5xUAS promoter for the six populations. (**C**) Heatmap plots of single-cell ATAC-seq reads from −200 to 200 bp around the TSS of the 5xUAS promoter. Reads are displayed in common logarithms. The counts were added to 0.1 before the logarithm conversion. The x-axis represents the distance from the TSS. The y-axis represents the cell index. Color represents the common logarithm of the reads. There are 384 cells for each time point. (**D–E**) For each cell, a read was further defined as the average over the 400 bp sequence. The mean reads (**D**) and CV of reads (**E**) over 384 cells plotted against time. Red dashed lines represent the dynamics of the PWM light induction.

The online version of this article includes the following source data for figure 4:

**Source data 1.** Source data for *Figure 4*.

openness) but fell drastically to 2.6 following just 150 min of illumination. The CV increased after each further 'off' phase and decreased after each additional 'on' phase to a lesser extent. These data indicated that the basal (dark) chromatin openness is very heterogeneous and that exposure of cells to a short period (150 min) of 100 μW/cm² light initiated transcription and reduced the heterogeneity of chromatin openness. Further increases in the chromatin openness and reduction in heterogeneity occurred after each consequent 'on' phase. The cells also maintained part of their epigenetic 'memory' during each 'off' period. Therefore, PWM induced an incremental reduction in the variability and an increase in the mean level of chromatin accessibility in each cycle of 'on' and 'off' phases. The periodic fluctuations in chromatin accessibility did not directly translate into single-cell protein mRuby dynamics, which did not exhibit such periodic fluctuations (*Figure 5—figure supplement 1*). The protein dynamics data indicated that the mean expression level reached a plateau at around 2000 min for PWM (*Figure 1I*). We postulate that chromatin openness will also eventually reach a plateau for both the 'on' and 'off' phases.

## PWM-induced oscillatory nuclear mRNA counts with a steady noise level

To connect the observed PWM-induced reduction in noise between the epigenetic and protein levels, we measured nuclear mRNA dynamics for single cells under AM and PWM light induction scenarios utilizing the MS2-MCP system (*Tutucci et al., 2018*). HeLa-AB2 cells, which exhibit PWM-induced noise reduction at 400 and 200 min period (*Figure 1—figure supplement 4B–C*), were further stably transfected with the LightOn mRNA imaging circuit (plasmid C2, shown in *Figure 1—figure supplement 1*) to obtain HeLa-ABC2 cells. We used live-cell spinning disk confocal microscopy to image the single mRNAs. When multiple MCP-tdTomato molecules are bound to an mRNA containing 24 MS2 hairpins, they form a punctum with a diameter of around 500 nm above the background fluorescence (illustrated as single mRNA puncta in the two nuclei in *Figure 5A*). We plotted the single-cell mRNA counts for all the tracked cells against time as heatmap images for AM (100 $\mu W/cm^2$), PWM (100 $\mu W/cm^2$, 200 min, 25% duty cycle), and AM (25 $\mu W/cm^2$), as shown in *Figure 5B–D*, respectively. For quantitative analysis, the means (*Figure 5E*) and CVs (*Figure 5F*) for single-nuclear mRNA numbers were plotted against time. For the AM with 100 $\mu W/cm^2$ (blue circles and line), the cells exhibited high variation at the beginning of the light-induction period that reduced over time, while the mean nuclear mRNA counts gradually increased and seemed to reach a high plateau after 1000 min (*Figure 5B, E– F*). For the AM with 25 $\mu W/cm^2$ (yellow circles and line), the cells exhibited more variation in mRNA numbers over time and a mean that approached a lower plateau (*Figure 5D–F*). These observations are consistent with the time course of flow cytometry analysis of mRuby protein (*Figure 1I-J*). For the PWM treatment, there is a clear pulsatile nuclear mRNA count with a period of 200 min, but the mean mRNA count appeared with a time delay of about 40 min after each 'on' time point, and as a population, these cells maintained a high mRNA count for the next 100 min or so before dropping down (*Figure 5C,E*). The variation in mRNA count was maintained at a more or less constant level that was intermediate between the levels for 25 and 100 $\mu W/cm^2$ AM light inductions (*Figure 5F*). There are also fewer cells without nuclear mRNA with PWM. Although sporadic nuclear mRNA counts might overshadow the periodicity of mRNA accumulation in the PWM case (*Figure 5C*), we observed oscillatory nuclear mRNA dynamics within the majority of cells by averaging over subpopulations of these cells with different mean RNA (*Figure 5G*). A period of 200 min, rather than 400 min, was chosen for PWM in the mRNA imaging experiment so that the periodicity of noisy nuclear mRNA dynamics could be recognized with more temporal repeats while noticeable noise reduction was still exhibited.

There are two noticeable phenomena concerning nuclear mRNA counts during PWM light induction. The heterogeneity of mRNA count remains steady over time and is consistently lower than that seen with AM of 25 $\mu W/cm^2$, no matter whether the cells were in the light 'on' or 'off' phases (*Figure 5F*). This suggests that the cells were likely to be alternating between high and low epigenetic states and not trapped in the bistable region (*Figure 2E*). The mean count exhibited a periodic oscillation. We did not observe periodic oscillation for the single-cell mRuby protein dynamics under PWM with 400 min duty cycle, shown in *Figure 5—figure supplement 1*, probably due in part to the longer half-life of protein.

Technical challenges that limited this type of long-term statistical analysis of single-cell mRNA dynamics remain. When compared to spinning disk confocal microscopy, light-sheet microscopy could increase the signal-to-noise ratio of these images, improve the fidelity of single mRNA detection, and ensure that an adequate number of repeating images are captured before significant photobleaching. Cell motility prevents the tracking of a sufficient number of cells over a long period, as most of the cells moved out of the imaging field at some time points. Further development of light-sheet microscopy technology is needed to provide the capability to image a large number of large optical fields while maintaining sufficient optical and temporal resolutions over a long period.

## Evidence of CBP/p300-induced heterogeneity in the regulation of endogenous genes

CBP/p300 is prominently implicated in regulating enhancer-dependent cell-type-specific gene expression (*Ogryzko et al., 1996*; *Weinert et al., 2018*). Compiled analysis of vast public ChIP-seq data from both human and mouse cell experiments suggests that there are more than 10,000 potential CBP/p300 target genes (*Oki et al., 2018*) (https://chip-atlas.org). Most of the target genes have

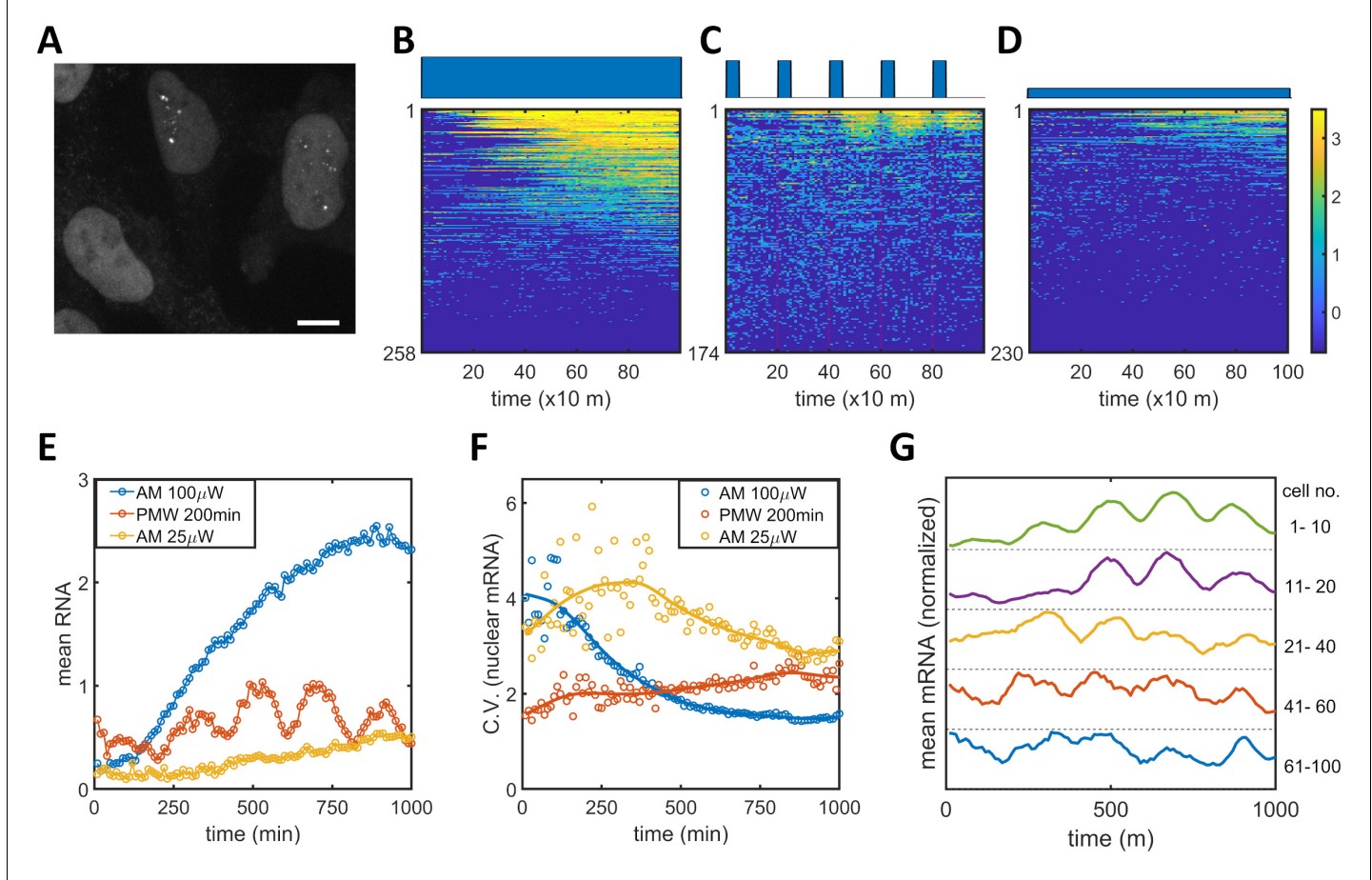

**Figure 5.** Single mRNA live-cell imaging for HeLa-ABC2 cells with AM and PWM. (**A**) A representative z-projection image at a specific time. The bright shapes with diameter around 10 µm represent the nuclei. The small brighter puncta (approximately 500 nm in diameter) in the nuclei represent nuclear mRNA molecules. The darker fluorescent signals are cytosolic fluorescent signals that are neglected in this analysis. The scale bar represents 10 µm. (**B–D**) The single-cell mRNA counts over time are detected and shown as heatmaps for AM with 100 µW/cm$^2$(**B**), for PWM with a maximum of 100 µW/cm$^2$, on-fraction of 0.25, and a period of 200 min (**C**), and for AM with 25 µW/cm$^2$ (**D**). The x-axis represents time with a unit of 10 min. The y-axis represents the cell index. The filled dark blue plots on top of the heatmap images represent the light induction schemes. Reads are displayed in natural logarithms. The counts were added to 0.5 before the logarithmic conversion. The percentages of cells without mRNA detection were 8%(**B**), 0.6%(**C**), and 20% (**D**). (**E–F**) The mean and CV for single-cell nuclear mRNA counts plotted against time for AM with 100 µW/cm$^2$ (blue), PWM with a period of 200 min and an on-fraction of 0.25 (red), and AM with 25 µW/cm$^2$ (yellow). (**G**) The means of single-cell mRNA counts for the PWM experiments in (**C**) over five subpopulations as indicated at the right of the plot. Each plot was normalized to its maximum. For each cell, a moving average of 60 min was applied before computing the average over subpopulations.

The online version of this article includes the following source data and figure supplement(s) for figure 5:

**Source data 1.** Source data for *Figure 5*.
**Figure supplement 1.** Single-cell mRuby dynamics with PWM.
**Figure supplement 1—source data 1.** Source data for *Figure 5—figure supplement 1*.
**Figure supplement 2.** Time-lapse images of representative nuclei.

complex mechanisms for gene regulation that overshadow the effects of the potential positive feedback loop formed by the transcriptional and histone acetylation functions of CBP/p300. Nevertheless, dual-functional CBP/p300 could increase heterogeneity in the expression of a subset of endogenous genes. To test whether CBP/p300 could upregulate heterogeneity in endogenous gene expression, we used A-485 and LMK-235 to inhibit CBP/p300 and HDAC4/5 and to disrupt the potential for positive feedback. In mES cells, the master transcriptional factors Nanog, Oct4 (*Pou5f1*), and Sox2 recruit CBP/p300 to facilitate ES cell-specific gene expression (*Chen et al., 2008*). mES cells that are cultured with LIF (but not with 2i cocktail) exhibit subpopulations with high and low expression levels of Rex1 (*Zfp42*) and Nanog, and with dynamic transitions between the two

states (*Singer et al., 2014*; *Toyooka et al., 2008*). To test whether CBP/p300 could contribute to bistability in mES cells, we constructed a Rex1-GFP/Oct4-mCherry knock-in cell line of mES D3 cells. This line exhibited bimodal Rex1-GFP expression, with 97% high and 3% low subpopulations, and unimodal Oct4-mCherry expression (*Figure 6A*). After treatment with LMK-235 and A-485 for two days, the expression of Rex1-GFP converged to an expression pattern centered between the high and low states (*Figure 6B*), similar to 5xUAS-mRuby expression in HeLa-AB1 cells treated with similar concentration of these inhibitors and AM light induction (*Figure 3D and F*). Although the global inhibition of CBP/p300 may also indirectly affect Rex1 expression, these results suggest that CBP/p300 may contribute to gene regulation in mES cells.

We further examined the effects of inhibitors of CBP/p300 and HDAC4/5 on global gene expression heterogeneity in HeLa-AB1 and F9-AB2 cells by performing scRNA-seq analysis. HeLa-AB1 cells were treated with either LMK-235 or A-485 and blue light for two days (A + L) or with light only (control). The library construction, sequencing, and data acquisition processes are described in detail in the Materials and methods section. As shown in *Figure 6C*, when the CV was plotted versus mean expression, most of the genes concentrated on a narrow band which could be fitted by a power function with an exponent of −0.48 (solid magenta line), similar to the predicted exponent of −0.5 of a Poisson distribution. Outlier genes that significantly deviated from this band (blue dots above the dashed red line) were further selected to assess the effects of A-485 and LMK-235. For the filtered gene set with lower CV values (less than two in control samples), there is a slight but statistically significant increase in mean CV from 1.30 to 1.45 (*Figure 6D* (magenta Pentastar) and E). On the other hand, for the filtered gene set with higher CV, there are many more genes for which large reductions in CV result from the inhibition of both CBP/p300 and HDAC4/5 (shown in *Figure 6D* (red triangles)). For these genes, the mean CV exhibits a large (3.49 to 2.55) and significant reduction in cells treated with the inhibitors. Within this set, genes that are targeted by CBP/p300 or by p65 show similar mean CV reduction (*Figure 6E*). There is much less deviation between the two biological repeats of control samples, especially for the high CV gene sets (*Figure 6D and F*). The reduction in CV resulting from inhibition of both CBP/p300 and HDAC4/5 can be visualized with narrower plots and computed from unique molecular identifier (UMI) counts, without or with the SAVER denoising process (*Huang et al., 2018*; *Luecken and Theis, 2019*; *Figure 6G*). Similar phenomena were observed with F9-AB2 cells (*Figure 6—figure supplement 1*). These results provide the first indirect evidence that CBP/p300 positively contributes to noise in global gene expression, especially for genes in human and mouse cells that show highly heterogenous expression. It could also contribute to the heterogeneous expression states in embryonic stem cells.

## Discussion

Synthetic inducible gene expression systems have been developed to study the function of genes in various cellular and physiological processes. The widely used systems allow researchers to induce signals such as by doxycycline or light, and have a high dynamic range of expression. They are essentially based on the simplest form of gene regulation involving one transcriptional activator. Nevertheless, they often exhibit huge noise in gene expression, especially in mammalian cells (CV as high as 5–10, shown in *Figure 1—figure supplement 4* and *Figure 2—figure supplement 1C*). The primary molecular mechanism that contributes to such a high level of noise has not been identified. In this study, we performed a quantitative characterization of gene expression noise in a light-induced expression circuit (*Wang et al., 2012*) in human and mouse cells under AM and PWM light-induction regimes. We found that PWM light controls noise in a period-dependent manner, enabling a phenomenological approach for independent modulation of expression level and gene expression dispersion by manipulation of mean light intensity and PWM period. This approach would support more precise design of cell fate control studies. Hardwired negative feedback circuits could reduce gene expression noise in mammalian cells but cannot independently modulate noise and expression (*Guinn and Balázsi, 2019*; *Nevozhay et al., 2013*). A recent study involved a more complex synthetic circuit with two orthogonal inducible systems arranged in series. These two inducible systems had unmatched noise properties, and were able to control the mean expression and noise independently by using different concentration combinations of two inducers (*Bonny et al., 2021*).

The time scales of PWM noise reduction and epigenetic regulation by HDAC4 and CBP/p300 overlap (*Bintu et al., 2016*; *Weinert et al., 2018*), prompting us to propose a working model of the

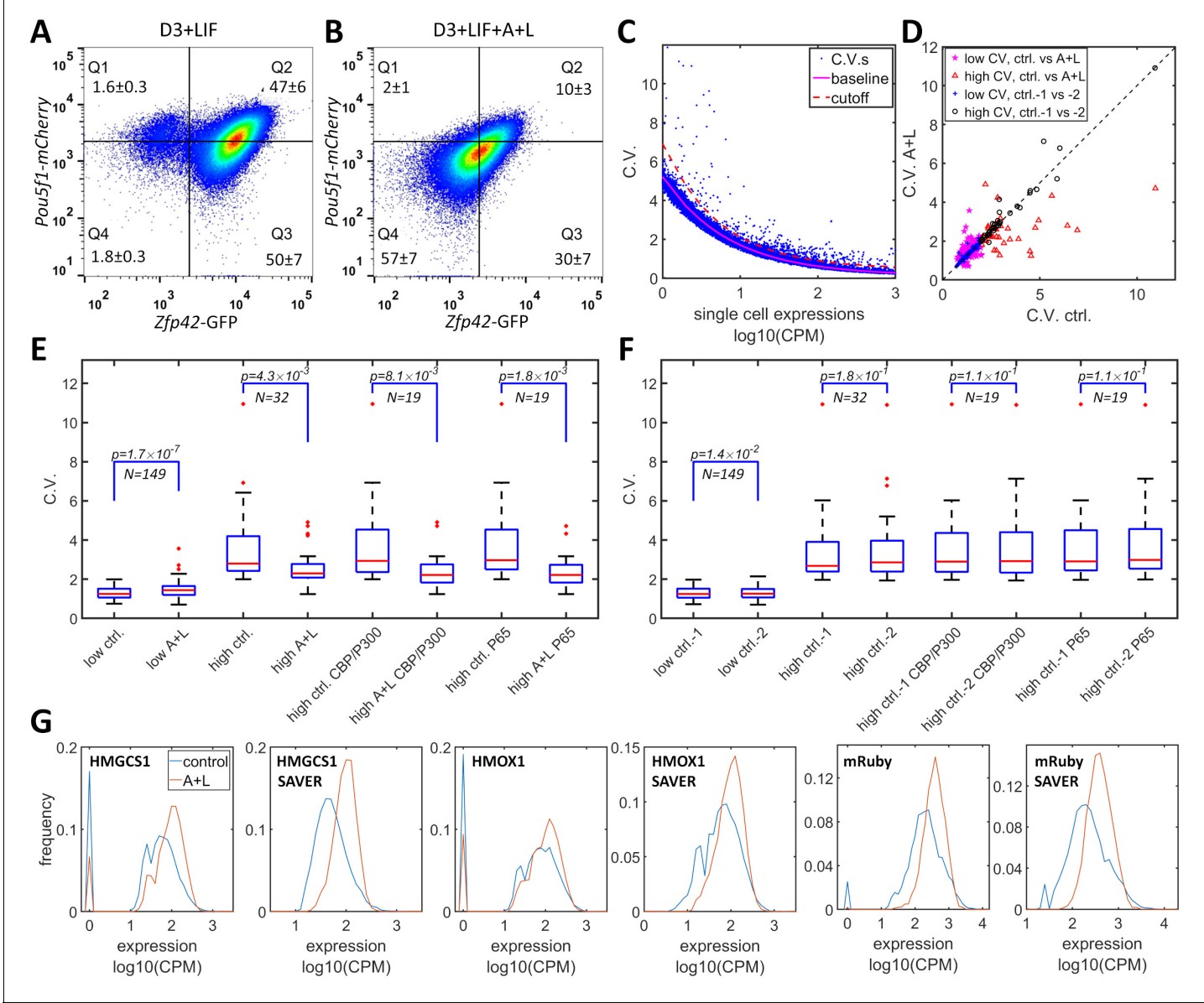

**Figure 6.** Simultaneous inhibition of CBP/p300 and HDAC4/5 reduces heterogeneity in endogenous gene expression in mES D3 and HeLa cells. (**A, B**) The density plots of Oct4-mCherry versus Rex1-GFP for a mouse ES D3 double knock-in cell line in regular mESC medium (**A**) or with the addition of 0.5 μM LMK-235 and 1 μM A-485 for two days (**B**). The mean +s.d. for the percentages of cells in Q1–Q4 were calculated from three (**A**) and six (**B**) biological repeats. (**C**) CV versus mean expression plot with scRNA-seq data from HeLa cells with or without 0.333 μM LMK and 1 μM A-485. The solid magenta line represents the fitted power function for all of the genes, which has an exponent of −0.48. The red dashed line represents the 3-sigma cutoff for noisy outlier genes. (**D**) The genes with CV versus gene expression above the cutoff (red line in **C**) and mean reads >5 CPM were selected for a plot comparing CV between HeLa cells with A485 +LMK-235 and control cells. Magenta stars and red triangles represent genes with low (<2.0) and high CV (≥2.0), respectively. Blue asterisks and black circles represent the CVs of the same gene sets from two biological replicates of control cells. (**E**) Statistical analysis of the CV of genes in cells treated with A485 +LMK-235 and in controls, comparing groups of genes with low CV, high CV, high CV, and positive for CBP/p300 or p65 ChIP. The boxes show the lower and upper quartiles; the whiskers show the minimum and maximal values excluding outliers; the line inside the box indicates the median; outliers (red dots) were calculated as values greater or lower than 1.5 times the interquartile range. The p-values shown in the figure were calculated using paired student t-tests. (**F**) Similar statistical analysis of CV between two biological control replicates for the same sets of genes. (**G**) Expression distribution of HMGCS1, HMOX1 and exogenous mRuby genes in control (blue) and A485 + LMK-235 groups (red), calculated from normalized data without or with the SAVER denoising algorithm. The peaks at '0' in the plots represent cells with zero reads.

The online version of this article includes the following source data and figure supplement(s) for figure 6:

**Source data 1.** Source data for *Figure 6*.

*Figure 6 continued on next page*

*Figure 6 continued*

**Figure supplement 1.** Simultaneous inhibition of CBP/p300 and HDAC4/5 reduces heterogeneity in endogenous gene expression in F9 cells.

**Figure supplement 1—source data 1.** Source data for *Figure 6—figure supplement 1*.

interplay between transcription and histone regulation. Our experimental system consists of a light-induced GAVPO dimer binding to a 5xUAS promoter in transiently open chromatin. The p65 activation domain of GAVPO recruits CBP/p300 to facilitate transcription initiation and histone acetylation. We propsose that the acetylated histones increase local chromatin accessibility and GAVPO binding. This postulated positive feedback loop is predicted to generate bimodal expression of genes under intermediate light induction. This mechanism was experimentally validated by disruption of expression bimodality using the selective inhibitor A485 to disrupt CBP/p300 HAT activity, specific disruption of GAVPO-CBP/p300 interaction by double point mutations in p65AD, and identification of low and high states of H3K27ac and of chromatin accessibility in the low and high expression cell populations. We further used single-cell ATAC-seq analysis and live-cell single mRNA imaging to illustrate the dynamic process of PWM noise reduction. scATAC-seq revealed that chromatin accessibility is highly variable in dark control conditions and was reduced after each cycle of light induction. Nevertheless, there was temporal fluctuation of the chromatin accessibility within each period. The noise level of single-cell nuclear mRNA counts for PWM remains lower than that for AM with intermediate light. Similar phenomena were observed at the protein level. In summary, PWM noise reduction by switching between the low and high states is established at the epigenetic level. The pulsatile chromatin accessibility and nuclear mRNA levels measured under PWM light induction were not detectable at the protein level, probably because the half-life of proteins are longer than those of mRNAs. Slow protein degradation might serve as a low-pass filter for mRNA oscillation and noise (*Wang et al., 2014*).

The mechanism that helps reduce temporal fluctuations might also limit the direct validation of bistability. Single-cell mRuby dynamics show that cells with similar GFP-GAVPO levels could reside in relatively stable high and low expression states (*Figure 2G,H*). The direct quantification of transitions between the high and low states is hindered by the long half-life and slow maturation time of mRuby proteins, and the limitation of the dynamic tracking length. We observed only minor hysteresis effects, probably for similar reasons (*Figure 2I,J*). Future experiments with bright fluorescent proteins or luciferases (*Suter et al., 2011*) that have significantly shorter half-lives could provide better direct assessments of the dynamic properties of bistability. It is also possible that additional epigenetic modifiers and transcription-related factors might contribute to more complex dynamic properties.

In nonlinear dynamics, chaotic oscillations can be stabilized at their unstable steady states using either the OGY feedback control theory (*Ott et al., 1990*) or periodical perturbations with specific frequencies (*Ding et al., 1994*). Recently, feedback and periodic control of aTc and IPTG were used to stabilize the unstable steady state in a genetic toggle switch in *Escherichia coli* (*Lugagne et al., 2017*). This particular scenario would lead to a decrease in mean gene expression with an increase in mean light intensity, illustrated as the black dashed line in *Figure 2E*, a phenomenon never observed in our experiments. *Benzinger and Khammash, 2018* have demonstrated noise reduction using PWM modulation of an EL222-based light-inducible gene expression system in *Saccharomyces cerevisiae*. They found that noise reduction is achieved by the PWM light working at the plateau regions of a single-valued sigmoidal active transcription factor to transcription response curve. In this scenario, transcriptional noise is mainly transferred from input noises, with bimodal expression being observed only with artificially inflated transcription factor heterogeneity in a non-isogenic cell population. Our PWM noise reduction mechanism is different. We found no sigmoidal GAVPO–mRuby response curve (*Figure 1G–H*), and the noise level under AM induction was too high to be explained by Benzinger and Khammash's hypothesis (*Figure 1D–H*). In addition, a narrow GFP-GAVPO input (CV <<0.05) could still generate bimodal mRuby expression and large noise (CV ~1.7), indicating that variation in the concentration of a single transcriptional factor led to multiple transcriptional states (*Figure 1—figure supplement 2C*). The T2A peptide linker ensured that tetR-GFP and GAVPO go through transcription and translation together but they may have separate post-translational processes. This might lead to different levels of noise. A recent study found that the intrinsic

variations between two 2A peptide-linked fluorescent proteins are minimal in mammalian cells (*Quarton et al., 2020*). A GFP–GAVPO fusion protein could provide direct quantitative analysis of the relation between GAVPO and mRuby, but it would reduce the activity of GAVPO and thus prevent sufficient mRuby expression for this study.

Our model suggests that the 5xUAS-mRuby at each locus has high and low expression states, and that the states at each locus are independent of each other. There are nine genomic insertion sites in HeLa-AB1 cells. The predicted nine high expression peaks are probably compounded into a broad 'high' peak, which is well separated from the 'low' peak, giving rise to an apparent overall bimodal distribution. Assuming that the nine sites have identical expression means and distributions at their 'high' and 'low' states, we emulated the combined mRuby expression distribution with increasing light intensities (*Figure 3—figure supplement 2*). The nine 'high' peaks formed an expanded peak and cannot be distinguished from each other due to the basal heterogeneity. The 'high' peaks are well-separated from the 'low' peak as the ratio of mRuby expression levels at the 'high' and 'low' states is sufficiently large. These emulated distributions are not so different from those of the experimental data (*Figure 1F*). The sizes of the inserted segments are too long (>2000 bp) to distinguish the epigenetic reads for each genomic insert in the current study. Further studies with newly developed single-cell epigenetic measurement techniques could help to assess this issue directly.

CBP/p300 proteins are reported to target large sets of genes and to regulate primarily enhancer-dependent cell-type-specific gene expression. We tested the consequences of potential disruption of this interaction on endogenous gene expression using both A485 and LMK-235. In a mES-D3 Rex1-GFP reporter cell line treated with both inhibitors, we found that the Rex1-high and Rex1-low subpopulations disappeared, resulting in the formation of a single population with intermediate Rex1 expression. In HeLa-AB1 cells, and to a lesser extent in F9-AB2 cells, scRNA-seq analysis suggested that the two inhibitors preferentially reduced the expression heterogeneity of high noise genes. These preliminary results indicate that the transcription-epigenetic interplay involving CBP/p300 could contribute to the highly heterogeneous expression of endogenous genes in mammalian cells.

In summary, we identified a transcription-epigenetic interplay involving p65AD and CBP/p300 that could lead to bimodal gene expression with high levels of noise. p65AD and VP16, which are used in most synthetic gene expression systems and in the regulation of many endogenous genes (*Gilbert et al., 2013*; *Gossen and Bujard, 1992*; *Gossen et al., 1995*; *Khalil et al., 2012*; *Wang et al., 2012*), are both reported to interact with CBP/p300 and other HATs (*Black et al., 2006*; *Gerritsen et al., 1997*; *Goodman and Smolik, 2000*; *Kim et al., 2012*; *Wang et al., 2000*). Abolishing this interaction could provide a general mechanism for the modulation of noise in gene expression in mammalian cells. CBP/p300 could also contribute to noise in the expression of some endogenous genes. We found that pulsed light induction with an extended period could reduce noise, presumably because it helps the cell to avoid staying in the bimodal regime. A picture of the dynamic process underlying this noise modulation, from chromatin accessibility to mRNA and protein levels, has been established using single-cell analysis. The noise reduction exhibits a dose-response to the period of PWM, which will enable independent modulation of mean gene expression and noise in future quantitative studies of cell-fate control.

## Materials and methods

### Plasmid constructions

Every version of the LightOn system is constructed with two plasmids: (1) an activator plasmid A coding for promoter-driven expression of the synthetic light-sensing transcriptional activator GAVPO, a fusion protein of the GAL4 DNA-binding domain, an engineered light-sensing VVD domain and the p65AD activation domain; and (2) a response plasmid B containing the LightOn promoter (5xUAS)-driven reporter genes encoding fluorescent proteins (*Figure 1A* and *Figure 1—figure supplement 1*). The plasmids containing the 5xUAS promoter and GAVPO (*Wang et al., 2012*) were gifts from Prof. Yi Yang. The plasmid for the tetR negative feedback circuit (similar to that described by *Nevozhay et al., 2013*) containing pCMV-tetO2-tetR-GFP-nuc was a gift from Prof. Jiandong Huang. The plasmids containing mRuby3 and mCardinal were gifts from Prof. Jun Chu. For

the HeLa cell line used in flow cytometry, live-cell imaging, and epigenetic assays, the transcriptional activator expression plasmid A1 was constructed by subcloning pCMV-tetO2-tetR-GFP-nuc with GAVPO into a plasmid with a PiggyBac transposon backbone (*Lu and Huang, 2014*) to form PB5-pCMV-tetO2-tetR-GFP-nuc-2A-GAVPO-2A-Zeo$^r$-βGpA-PB3. The LightOn plasmid B1 was constructed by assembling 5xUAS, mRuby3, and subcloning it into a plasmid with a PiggyBac transposon backbone (*Lu and Huang, 2014*) to form PB5-5xUAS-mRuby-nuc-2A-Bla$^r$-βGpA-PB3. For F9 and HeLa cell transformation, the transcriptional activator plasmid A2 was constructed by subcloning GAVPO into another PiggyBac backbone to form PB5-pCAG- GAVPO-2A-Zeo$^r$-βGpA-PB3. The LightOn plasmid B2 (or B3) was constructed by subcloning 5xUAS, GFP-nuc (or mCardinal-nuc) into the PiggyBac backbone plasmid to form PB5-5xUAS-GFP-nuc-2A-Bla$^r$-βGpA-PB3 (or PB5-5xUAS-mCardinal-nuc-2A-Bla$^r$-βGpA-PB3).

To reduce the half-life of the reporter fluorescent protein and to increase its temporal response, we fused amino acids 422–461 of the degradation domain of mouse ornithine decarboxylase (MODC) (a PEST sequence) (*Li et al., 1998*) to the C-terminal end of mCardinal to form plasmid B4 (PB5-5xUAS-mCardinal-PEST-2A-Bla$^r$-βGpA-PB3). To abrogate the interaction between the transactivator GAVPO and CBP/p300, we generated L449A/F473A double mutations in the activation domain of p65 on plasmid A2 to obtain plasmid A3 (*Mukherjee et al., 2013*). For the HeLa cell line used in mRNA imaging, in addition to transcriptional activator plasmid A2 and LightOn plasmid B2, an additional plasmid, C2, was constructed by subcloning 5xUAS-tagBFP-nuc with 24 MS2 hairpin repeats (synthesized by Wuxi Qinglan Biotech), the SV40 promoter, and MCP-tdTomato-nuc (synthesized by Wuxi Qinglan Biotech) into a PiggyBac backbone plasmid to form PB5-pSV40- MCP-tdTomato-nuc-2A-Puro$^r$ -SV40pA-HS4-5xUAS-BFP-nuc-24xMS2-βGpA-PB3. In all of these constructs, 2A stands for a 2A 'self-cleaving' peptide T2A, derived from Thosea asigna virus 2A, which enables the same levels of transcription and translation for the two proteins. The nuc tag consists of three tandem repeats of the SV40 nuclear localization sequence, which facilitate the localization of the protein in the nucleus. The UAS is the specific GAL4-binding element. Puro$^r$, Zeo$^r$, and Bla$^r$ are resistance genes for the antibiotics puromycin, zeocin, and blasticidin, respectively. HS4 is the Chicken hypersensitive site 4 (cHS4) insulator. The components of the MS2-MCP system (MS2 RNA loop and MS2 coat protein) were adapted from *Tutucci et al., 2018*. To modify the transactivator rtTA in the classic tetON system, we replaced its VP16 domain with p65AD to form plasmid A4 (PB5-pCAG-rtetR-p65AD-2A-Zeo$^r$-βGpA-PB3). A response plasmid B5 for the tetON system was constructed using the 5xUAS promoter in plasmid B2 with TRE-3G promoter containing seven tetO sites. The first six tetO sites were further removed to construct plasmid B6.

To generate the knock-in Rex1-GFP and Oct4-mCherry mES cell line, we constructed two donor plasmids D1 and D2. For construction of the Rex1-GFP knock-in donor plasmid D1, a T2A-GFP-nuc cassette was in-frame fused with a 5' 1000 bp homologous arm, including the C-terminus of the Rex1 gene, and a 1000 bp 3' arm including the 3'UTR sequence of the Rex1 (Zfp42) gene. For the Oct4-mCherry knock-in donor plasmid D2, an ATG-mCherry-linker cassette was in-frame fused with an 800 bp 3' arm, including the N-terminus of the Oct4 coding sequence, and an 800 bp 5' arm, including the 5'UTR sequence of Oct4. The gRNA sequences for Rex1 and Oct4, TCCTAACCCACG-CAAAGGCC and CAGGTGTCCAGCCATGGGGA, respectively, were cloned into a pX330 gRNA-Cas9 plasmid (Zhang Feng lab) to obtain Oct4-gRNA-Cas9 and Rex1-gRNA-Cas9 plasmids. All other components, unless otherwise specified, were adapted from *Lu and Huang, 2014*. All of the plasmid constructions were confirmed by Sanger sequencing. The sequences of the plasmids and oligos used for the construction of plasmids will be provided upon request.

## Construction of cell lines

To generate HeLa cells that are stably transfected with GFP-GAVPO and LightOn-mRuby, we cotransfected HeLa cells with 400 ng plasmid A1, 400 ng plasmid B1, and 200 ng PGK-transposase plasmids using Lipofectamine 3000 (Catalog no. L3000-015; Invitrogen) following the manufacturer's protocol. The cells were transfected in one well of a 24-well plate and selected with 100 µg/mL zeocin (Catalog no. R25001; Gibco) for six days, starting one day after transfection. The medium was replaced with DMEM supplemented with 10% FBS and 1 µg/mL doxycycline (Catalog no. D9891; Sigma), and the cells were illuminated with blue light (100 µW/cm$^2$) for 48 hr. Single cells expressing both EGFP and mRuby were sorted into 96-well plates using BD FACSAria SORP. Each clone was verified using flow cytometry two weeks later. One clone, named HeLa-AB1, was chosen for further

analysis. Similarly, F9 and HeLa cells were stably transfected with 400 ng plasmid A2, 400 ng plasmid B2, and 200 ng PGK-transposase plasmids. The cells were illuminated with blue light for two days before sorting single cells that expressed GFP. Single clones of F9 cells were analyzed ten days after sorting. One F9 clone, named F9-AB2, was chosen for further analysis. Another HeLa clone, named HeLa-AB2, was further transfected with the mRNA imaging plasmid C2 and PGK-transposase plasmid, and was selected by culturing with 1.5 µg/mL puromycin (Catalog no. A1113803; Gibco) for three days. Cells were illuminated for two days with blue light and sorted by positive GFP, tdTomato, and BFP expression. One clone, named HeLa-ABC2, was chosen for mRNA imaging. To generate HeLa cells that were stably transfected with pTet-GFP, we transfected HeLa cells with a pBX-123 plasmid (*Lu and Huang, 2014*) and PGK-transposase plasmid, and selected the stable clones with 1.5 µg/mL puromycin (Catalog no. A1113803; Gibco) for three days. One clone, named HeLa-Tet-On, was chosen for further analysis. HeLa cells were stably transfected with 400 ng plasmid A3, 400 ng plasmid B2, and 200 ng PGK-transposase plasmid, selected with 100 µg/mL zeocin, and illuminated with blue light (100 µW/cm$^2$) for six days. GFP-positive single cells were then sorted, expanded, and genotyped to obtain the HeLa-A3B2 cell line. HeLa cells were stably transfected with 400 ng plasmid A4, 400 ng plasmid B5 (or B6), and 200 ng PGK-transposase plasmids, selected with 100 µg/mL zeocin, induced with 1000 ng/mL doxycycline for six days. GFP-positive single cells were sorted, expanded, and genotyped to obtain the HeLa-A4B5 and HeLa-A4B6 cell lines. In addition, HeLa-AB1 cells were further transfected with 400 ng plasmid B4 and 200 ng PGK-transposase plasmid, and illuminated with blue light (100 µW/cm$^2$) for six days. Single cells with mCardinal, GFP, and mRuby signals were then sorted expanded, and genotyped to obtain the HeLa-A1B1B4 cell line.

In addition, HeLa-AB1 cells were transfected with 200 ng PGK-transposase plasmid and a reduced amount of plasmid B3 (40 ng) to obtain 5xUAS-mCardinal clones with fewer integration sites, thereby generating HeLa-A1B1B3 clones. The integration sites were quantified by qPCR of genomic DNA with primers for mCardinal 5'-cttcatcaaccacacccag-3' and 5'- ccatgtcgcatctgccttc-3', and TB Green Premix Ex Taq II (Catalog No. RR820A; Takara). Genomic DNA was purified using the Pure-Link Genomic DNA Mini Kit (K1820-01, Invitrogen), and 100 ng per reaction was used. A standard curve was generated by serial dilution of plasmid B3 using the same concentration of wild-type HeLa genomic DNA. The total genome size of HeLa cells was estimated to be approximately 1.3-fold that of the normal human genome (*Adey et al., 2013*).

## Constructions of knock-in mouse ES cell lines

To generate knock-in Rex1-GFP and Oct4-mCherry mES cell lines, mES cell line D3 (a gift from Prof. Jianbo Yue) was cultured on a 0.1% gelatin-coated 24-well plate with mESC medium (DMEM containing 15% ES FBS, 1000 U/ml LIF, 1X GlutaMax, 1X non-essential amino acid (NEAA) and 0.1 mM 2-mercaptoethanol) until it reached 50% confluence. D3 cells were transfected with 500 ng Oct4-mCherry donor plasmid (D1) and 500 ng Oct4-gRNA-Cas9 plasmids with Lipofectamine 3000 and diluted one day later into a 6-well plate, in which they were maintained for four days in the mESC medium containing 2i cocktail (3 µM CHIR99021 and 1 µM PD0325901). Single cells expressing mCherry were sorted, expanded, and genotyped by PCR and sequencing. One validated Oct4-mCherry clone was used to repeat the knock-in experiments with the Rex1-GFP donor plasmid (D2) and the Rex-gRNA-Cas9 plasmid.

## Light induction of expression

HeLa-AB1 cells and their derivatives were plated in an ibidi 24-well µ-plate at a density of $3 \times 10^4$ cells/well. The medium was supplemented with 1 µg/mL doxycycline (dox) on day −1. On day 0, the dox-containing medium was replenished, and the µ-plate was mounted on a customized 24-channel illumination apparatus placed in a Thermo Heracell 150i CO2 incubator. A water-cooling pump was turned on to ensure that cells were cultured at 37°C. A CSV file containing the designed illumination intensities and dynamics for the 24 wells was loaded into the custom Python code to control the apparatus for 48 hr unless otherwise specified. The procedure was similar for F9-AB2, HeLa-AB2 cells, and other light-inducible cell lines, except that doxycycline was not added.

## Finding PWM parameters to specify mean expression levels and distribution spreading

We recorded flow cytometry data for HeLa-AB1 cells, with AM (period = 0) and PWM with different periods (see *Figure 1—figure supplement 3A–F*), calculated the mean mRuby emission to mean light intensity curve, and distribution spreading (defined as the ratio of mRuby emission at the 90th and 10th percentiles) to mean light intensity. We then fitted the data using a cubic smoothing spline function (csaps) with MATLAB 2018b to generate response-surface plots for the common logarithms of mean value and distribution spreading against period and mean light intensity (*Figure 1—figure supplement 3H–I*). We chose the distribution spreading ratio as read-out instead of the coefficient of variation, because this ratio provides a more intuitive picture of how diverse the gene expressions were and is not affected by instrumentation settings.

## Experiments with inhibitors of epigenetic regulators

HeLa-AB1 cells were plated in an ibidi μ-plate 24-well plate at a density of $3 \times 10^4$ cells/well. The medium was supplemented with 1 μg/mL doxycycline on day −1. On day 0, the medium was supplemented with 0.333 μM (unless otherwise specified) LMK-235 (Catalog no. S7569; SelleckChem), 0.333 μM Vorinostat (Catalog no. S1047; SelleckChem), 0.333 μM PF-06726304 (Catalog no. S8494; SelleckChem), or 1 μM of A-485 (Catalog no. S8740; SelleckChem). On day 0, the μ-plate was mounted on the 24-channel illumination apparatus located in the $CO_2$ incubator and illuminated for two days before flow cytometry analysis.

For Oct4-mCherry-Rex1-GFP mES cells, the 2i cocktail was removed from the mESC medium for at least four days earlier to establish heterogenous mES cell culture with Rex1-GFP-high and Rex1-GFP-low populations. Cells were then treated with 0.5 μM LMK-235 and 1 μM A-485 for two days before performing flow cytometry analysis.

## Flow cytometry analysis

Cells in one well of a 24-well plate were monodispersed using 125 μL of 1X TrypLE Express Enzyme (Gibco), neutralized with 250 μL of culture medium, filtered through a 40 μm cell strainer (BD Falcon) to remove clumps, and analyzed using a Beckman Cytoflex S cytometer (Brea, US) equipped with 405/488/561/640 nm lasers at high speed (60 μL/min) for 2 min per sample. The remaining samples were kept on ice. To ensure the reproducibility of the measurements, we used fluorescent calibration beads (Sphero Rainbow calibration beads six peaks; Catalog no. RCP-60–5; Spherotech) to adjust the instrumentation parameters.

## Analysis of flow cytometry data

The flow cytometry data were gated for single cells and exported as CSV files using FlowJo X (Ashland, US). A custom MATLAB (Mathworks, Natick, US) script was used to load the CSV files for quantitative analysis, including mean, CV, distribution of expression, etc. The relevant fluorescent channel was normalized to the mean value of the fifth peak of the Rainbow calibration beads. There was no detectable spillover between the GFP (FITC-A), mRuby (ECD-A), mCardinal (APC-A), and BFP (PB450A) channels, so compensation was not needed.

## Predictions of mRuby from GFP-GAVPO with single-valued propagation functions

The expression of two fluorescent proteins linked by a 2A peptide are well-correlated at the single-cell level (*Quarton et al., 2020*). Thus, single-cell tetR-GFP expression is a close approximation of GAVPO expression as these two genes are linked with a T2A peptide in HeLa-AB1 cells. At a specific AM light induction, the activated GAVPO concentration is estimated to be proportional to total GAVPO concentration in single cells and limited by the expression level of GAVPO. Therefore, single-cell tetR-GFP expression is a reasonable surrogate for single-cell active GAVPO concentration at a specific AM light condition. To estimate the empirical propagation function of GAVPO to mRuby, we extracted single-cell GFP and mRuby expression and sorted the cell population by rising GFP expression. The total population of cells (*Figure 1G*) was divided into 20 subpopulations with incremental mean GFP values and narrow ranges. An empirical function (red dashed line in the center panel of *Figure 1H*) was fitted to the mean mRuby values vs. mean GFP values for the

subpopulations (red circles in the center panel of *Figure 1H*). A hypothetical propagation function with high cooperativity was formulated with a Hill function with a Hill coefficient of 8 (black dashed line in the center panel of *Figure 1H*). A phenomenological model was then used to compute a single-cell mRuby value from the single-cell GFP-GAVPO value using a single-valued propagation function *f*:

$$R = f(G) + noise \tag{1}$$

in which the function *f* can be treated as the fraction of pre-initiation complex recruitment to the promoter, and the 'noise' represents the stochastic contributions from transcription, translation, etc. We added a lognormal noise with an amplitude of 0.2 and proportional to the predicted mean mRuby expression, contributing an additional 0.48 to the CV. The detailed algorithm is described in the MATLAB code in Appendix 2.

## Live-cell confocal imaging of single mRNAs with blue light illumination

HeLa-ABC2 cells were plated on 3.5 cm glass-bottom FluoroDishes (FD35PDL, MPI) at a density of 3 $\times$ 10$^5$ cells/dish one day in advance. Immediately before imaging, the medium was supplemented with 10 mM HEPES, pH 7.4, and 1% penicillin/streptomycin, and the dish was placed on an Andor CSU-X1 spinning disk confocal microscope with a Hamamatsu Orca-Fusion sCMOS camera, Nikon 60X NA1.49 TIRF objective, and Metamorph software. The blue-light illumination was set up by inserting an Arduino-controlled shutter (Daheng Optics GCI-7103) and a blue band pass filter (Omega 465/40DF) in the light path of the bright field illumination. The bright field illumination was adjusted to a blue light spot of 7 mm diameter centered at the bottom of the 35 mm dish, with light intensities of 25 or 100 µW/cm$^2$. Multiple XY positions within a 3 mm region were chosen, so that every cell in the optical field was always within the 7 mm blue illumination circle and the positions were separated enough to avoid overlapping in excitation and photobleaching. At each XY-position, 20 z-slices with a spacing of 0.5 µm were recorded with 200 ms exposure per slice. A Prior piezo-z stage was used to ensure z precision. Each XY position was imaged every 10 min for 1000 min at 37˚ C and 5% CO$_2$. A 561 nm laser was chosen for excitation to ensure the detection of single mRNA punctae while minimizing photobleaching over 100 3-D imaging loops. A 617/73 emission filter was used to collect tdTomato fluorescence. The AM light of 100 or 25 µW/cm$^2$ or PWM with 200 min period, 25% on duty, and 100 µW/cm$^2$ max intensity was set up and started immediately before the start of the live-cell confocal imaging. The period of 200 min was chosen over 400 min to enable multiple cycles of light induction without significant photobleaching. This periodicity also significantly reduces noise over the AM (*Figure 1—figure supplement 4B–C*).

## Processing of single mRNA imaging

MCP-tdTomato-nuc fluorescence is mostly localized inside the nucleus, which enabled us to segment and track individual nuclei, and to detect mRNA in a single channel (*Figure 5A*). Each optical field was captured in a 4D XYZT image with 1300 pixels $\times$ 1900 pixels $\times$ 20 slices $\times$ 100 timepoints. With a camera pixel size of 6.5 µm, a 60X NA1.49 TIRF objective, thickness of 0.5 µm per z-slice, and 10 min imaging frequency, with each 4D image recorded 140 $\times$ 205 $\times$ 9.5 µm$^3$ over 1000 min. The cells moved significantly during this 1000 min period, and most of the nuclei moved outside of the imaging boundary at one time or another. To find all nuclei that stayed entirely inside the boundary, we performed nuclear segmentation and tracking before using Imaris 9.4 to identify and count mRNAs. The first step in this process is the generation of a z-axis maximum projection. A medium filter (window size = 5) and Laplacian of Gaussian filter (sigma = 40) were applied sequentially to projected XYT images, and a threshold was chosen to binarize the images and identify the centroids of the nuclei. Connected components were searched in these binary images over time to track the nuclei that remained inside the imaging boundary. In addition, for each z-projection image, using the identified nuclei centroids as seeds, a watershed algorithm was applied to find the 'basin' occupied by each cell at every time point. For all the tracked nuclei, the entire cell region was cropped from the original XYZT images based on its 'basin' at each time point, and re-centered to generate a 512 $\times$ 512 $\times$ 20 $\times$ 100 box. The nucleus was finally segmented using Otsu's algorithm in the cropped image. These XYZT images were normalized to the mean nuclear fluorescent intensity to compensate for MCP-tdTomato-nuc expression heterogeneity and possible photobleaching, as

Imaris doesn't incorporate adaptive thresholding. The single mRNAs were identified using Imaris's spot model with an estimated x-y diameter of 0.54 μm and z diameter of 1.5 μm. All of the code scripts were written in Julia 1.4.1 using the JuliaImages v0.20 package.

## Live-cell wide-field fluorescent imaging, tracking, and quantification of single-cell dynamics with blue-light illumination

A customized 6-channel LED illumination apparatus was built to implement dynamic blue-light induction without interference from live-cell fluorescent imaging (*Appendix 1—figure 3*). HeLa-AB1 cells were plated in an ibidi μ-plate 24-well plate at a density of $1 \times 10^4$ cells/well, and the medium was supplemented with 1 μg/mL doxycycline on day −1. On day 0, the cells were placed on a Ti-E microscope with a Lumencor SpectraX solid-state light source, a Nikon Plan Apo 20 × 0.75 objective, a Andor Zyla 5.5 sCMOS camera, a Nikon motorized stage, and PFS autofocus systems. The cells were maintained at 37°C and 5% $CO_2$ in a customized environmental chamber with blue-light illumination from the top. GFP (filter set: excitation FF01-485/20, emission FF01-525/30, dichroic Di01-R405/488/561/635; Semrock) and mRuby (filter set: excitation FF01-560/25, emission FF01-590/20, dichroic Di01-R405/488/561/635; Semrock) channels were imaged every 90 min using Nikon NIS software. The illumination nonuniformity was corrected by normalization against images of a well with media only. The nuclei of cells were segmented and binarized using the GFP images by searching for and filtering local maxima in the Gaussian blurred image as seeds, then applying the watershed algorithm to split individual cells. Each nucleus was extracted by thresholding with Otsu's algorithm in each split region. The tracking was carried out in a reverse time sequence. In any pair of temporally adjacent images, connected cells were identified by closest proximity within a minimal search range of 20 μm. The entire track for each cell was validated with the criteria of continuity in GFP expression (less than 30% relative change, except for cell division events) and manually inspected. For quantification, the local background signal (median signal of nearby 400 × 400 pixels that don't contain any nucleus) was subtracted from the signal of a segmented nucleus and the resulting signal was integrated. To represent continuous expression dynamics, we normalized single-cell expression dynamics with the cell-cycle state for that cell (*Figure 2—figure supplement 3C*). All the code scripts were written either in Julia 1.4.1 using the JuliaImages v0.20 package or in MATLAB 2018b.

## Whole-genome sequencing of HeLa-AB1 cells

The whole-genome sequencing (50X) of the HeLa-AB1 clone was performed by GENEWIZ (Suzhou, China). The reads were processed to call potential plasmid insertion sites by GENEWIZ (Suzhou, China). The reads were trimmed with cutadapt 1.9.1 to remove dual ends with a base quality cutoff of 20 and to remove adapters. The trimmed reads were filtered with cutadapt 1.9.1 to discard reads containing more than 10% N bases and reads that were shorter than 75 bp. The filtered reads were mapped to the reference genome (hg19) and plasmid insertion sequences using bwa 0.7.12-r1039. The alignments were sorted by leftmost coordinates using samtools 1.6. The duplicated reads in sorted alignments were marked with the MarkDuplicates function of Picard 2.2.1. The break sites between the hg19 and plasmid were called from the alignments using VIFI 0.2.13. These sites were potential plasmid insertion sites in hg19 coordinates. PCR and Sanger sequencing were used to confirm the plasmid insertion sites. We confirmed nine insertion sites for plasmid B1 and five for plasmid A1 (*Figure 3—figure supplement 1B*). The plasmid insertion sequences were inserted into the human reference genome (hg19) according to the validated insertion sites to build the single cloned cell line reference genome (HeLa-AB1) using a custom Python3 script. Every insertion site was updated after the front sites in the same chromosome had been inserted.

## Chromatin immunoprecipitation-sequencing (ChIP-seq) and ATAC-seq sample preparation

HeLa-AB1 cells were plated in a 10 cm dish at a density of $1.5 \times 10^6$ cells/dish. The medium was supplemented with 1 μg/mL doxycycline on day −1 to induce GAVPO expression. On day 0, cells were illuminated by 25 μW/cm$^2$ blue light for 24 hr to generate large mRuby distribution dispersion (expression heterogeneity). Cells were monodispersed and filtered through a strainer to remove clumps. FACS Aria SORP was used to sort cells into high and low mRuby populations. Cells were kept on chilled water before and during sorting. Single cells were selected on the basis of their side

and forward scattering properties. Manual gates were imposed on the mRuby fluorescence to collect approximately 20% of the lowest and highest mRuby-expression cells (*Figure 3I*). For each population, $5 \times 10^6$ cells were collected. A fraction of the cells ($\sim2\times10^5$) were cryopreserved with DMEM supplemented with 40% FBS (Gibco) and 10% DMSO (Catalog no. D8418; Sigma) for ATAC-seq analysis (GENEWITZ, Suzhou, China). The rest of the cells were washed twice in cold PBS buffer and cross-linked with 1% formaldehyde for 10 min at room temperature, and then quenched by adding glycine (125 mM final concentration) for subsequent ChIP-seq experiments.

## ChIP-seq

The following ChIP-seq experiment was performed at IGENBOOK Biotech (Wuhan, China). Samples were lysed, and chromatins was sonicated to obtain soluble sheared chromatin (average DNA length of 200–500 bp). Twenty microliters of chromatin were saved at –20°C for input DNA, and 100 µL chromatin was used for immunoprecipitation. The antibody used for H3K27me3 was ab4729 (Abcam). Ten micrograms of antibody were used in the immunoprecipitation reactions at 4°C overnight. The next day, 30 µL of protein-A beads were added, and the samples were further incubated for three hours. The beads were then washed once with 20 mM Tris/HCL (pH 8.1), 50 mM NaCl, 2 mM EDTA, 1% Triton X-100 and 0.1% SDS, then twice with 10 mM Tris/HCL (pH 8.1), 250 mM LiCl, 1 mM EDTA, 1% NP-40 and 1% deoxycholic acid; and twice with $1 \times$ TE buffer (10 mM Tris-Cl at pH 7.5, 1 mM EDTA). Bound material was then eluted from the beads in 300 µL of elution buffer (100 mM $NaHCO_3$, 1% SDS), treated first with RNase A (final concentration 8 µg/mL) for six hours at 65°C and then with proteinase K (final concentration 345 µg/mL) overnight at 45°C. Immunoprecipitated DNA was used to construct sequencing libraries following the protocol provided by the NEXTFLEX ChIP-Seq Library Prep Kit for Illumina Sequencing (NOVA-514120, PerkinElmer Applied Genomics) and sequenced on Illumina Xten with the PE150 method.

## ATAC-seq

The following ATAC-seq experiment was performed at GENWITZ (Suzhou, China). The cell pellet was resuspended in 50 µL of tagmentation mix (12.5 µL 4X THS-seq TD buffer, 5 µL 0.1% digitonin (Catalog no. G9441; Promega), 5 µL TTE Mix V50 (Catalog no. TD501; Vazyme), and 27.5 µl $H_2O$). The tagmentation reaction was performed on a thermomixer (Eppendorf 5384000039) at 800 rpm, 37°C, 30 min. The tagmented DNA was purified following Qiagen MinElute PCR purification kit protocols. The library was constructed using 10 µL purified tagmented DNA, 2.5 µL S5XX and 2.5 µL N7XX indexing primers, 25 µL of NEBNext 2X PCR MasterMix and 10 µL $H_2O$, amplified at 72°C for 5 min, 98°C for 1 min, then [98°C for 10 s, 63°C for 30 s, 72°C for 20 s] $\times$ 12. The product was purified using the SPRI beads method (Beckman Coulter). Sequencing was performed on an Illumina HiSeq with the PE150 method.

## Single-cell ATAC-seq

The period of 600 min was chosen for PWM light induction as it reduces noise and was better suited for covering chromatin dynamics studies from 0 to 1 day (*Figure 4A*). Briefly, 50,000 cells were centrifuged at 500 $\times$ g, 4°C, 5 min. Cell pellets were resuspended in 50 µL tagmentation mix (33 mM Tris-acetate, pH 7.8, 66 mM potassium acetate, 10 mM magnesium acetate, 16% dimethylformamide (DMF), 0.01% digitonin, and 5 µL of Tn5 from the Nextera kit from Illumina (Cat. No. FC-121–1030)). The tagmentation reaction was performed on a thermomixer at 800 rpm, 37°C, 30 min. The reaction was then stopped by adding an equal volume (50 µL) of tagmentation stop buffer (10 mM Tris-HCl, pH 8.0, 20 mM EDTA, pH 8.0) and left on ice for 10 min. A volume of 200 µL 1X DPBS with 0.5% BSA was added, and the nuclei suspension was transferred to a FACS tube. Tagmented single nuclei were sorted on the basis of GFP signal (nuclear-localized tetR-GFP-nuc) using BD FACSAria SORP in one prepared 384-well plate containing 4 µL lysis buffers (50 mM Tris, pH 8.0, 50 mM NaCl, 20 ug/ml Proteinase K, 0.2% SDS, and 10 µM S5xx/N7xx Nextera index primer mix (5 µM each)). After sorting, the plate was briefly centrifuged and incubated at 65°C for 30 min. Then, 4 µL 10% Tween-20, 2 µL $H_2O$ and 10 µL NEBNext High-Fidelity 2 $\times$ PCR Master Mix were added to each well sequentially. Libraries were amplified with 72°C for 5 min, 98°C for 5 min, [98°C for 10 s, 63°C for 30 s, 72°C for 20 s] $\times$ 18. Finally, all reactions were pooled together and purified with a PCR minElute purification

column (Qiagen). Libraries were sequenced at HaploX Biotech (Shenzhen, China) with a HiseqX machine with 40G per one 384-well plate.

## Single-cell RNA-seq library preparation

HeLa-AB1 and F9-AB2 cells were seeded into an ibidi 24-well µ-plate at 30,000 cells per well one day before the experiments. At day 0, two wells (A2, A3, control) and one well of HeLa-AB1 cells (C2, inhibitors) were replenished with regular medium or with regular medium containing 0.333 µM LMK-235 and 1 µM A-485, respectively, and illuminated with blue lights for two days at 21 and 9 µW/cm$^2$, respectively. At the same time, two wells (A1, B1, control) and one well (C1, inhibitors) of F9-AB2 cells with regular medium and regular medium containing 1 µM LMK-235 and 1 µM A-485, respectively, were cultured in the dark for 2 days. F9-AB2 cells (three samples named A1, B1 and C1, respectively) and HeLa-AB1 cells (three samples named A2, A3 and C2, respectively) in an ibidi 24-well µ-plate were washed three times with 500 µL DMEM, followed by incubation in 125 µL TrypLE Express Enzyme at 37 °C for 5 min. This enzyme was deactivated by 250 µL DMEM + 10% FBS, and the cells were transferred into 1.5 mL centrifuge tubes and centrifuged at 200 × g, 4 °C for 5 min. The resulting cell pellets were washed with 1 mL PBS containing 0.1% w/v BSA. After discarding as much PBS as possible and resuspending the cell pellets with an additional 100 µL PBS containing 0.1% BSA. The cells were filtered through the filter cap of the flow tube (Catalog no. 352235; BD Falcon). Aliquots of 10 µL of filtered cells were mixed with equal volumes of trypan blue solution for cell counting with the Countstar analyzer (RuiYu Biotech, Shanghai, China). Finally, the human cells (HeLa-AB1 A2, A3, and B2) and mouse cells (F9-AB2, A1, B1, and C1) were mixed at a 1:1 ratio to form three samples (A1/A2, B1/A3, C1/C2). Subsequently, according to the 10 × Genomics user guide, scRNA-seq libraries were constructed in three steps: GEM Generation and Barcoding, Post GEM-RT Cleanup and cDNA Amplification, and 3' Gene Expression Library Construction. In the first step, to obtain a final recovery of 10,000 cells/sample, 16,500 cells/sample were loaded into the 10 × machine (Chromium Next GEM Single Cell 3' Library and Gel Bead Kit v3.1). In the third step, after the scRNA-seq libraries were constructed, each library was split into two sets. One set was sequenced on the HiseqX system at HaploX Biotech, whereas the other was sequenced on the MGI2000 platform at BGI.

## Analysis of ChIP-seq and ATAC-seq data

Reads were preprocessed using the fastp 0.20.0 software, with the default setting and correct low-quality mismatched base pairs in overlapping regions of paired-end reads. The preprocessed reads were mapped to the single cloned cell line reference genome (HeLa-AB1) using bowite2 2.3.5.1. The alignments were preprocessed with samtools 1.9 and then deduplicated using the MarkDuplicates function of Picard 2.22.2. The deduplicated alignments were filtered using samtools 1.9 with SAM flag (-F 1804 f 2) to select reads mapped in the proper pair. These reads were filtered using custom bash code to remove the reads that were mapped to mitochondria. For population sequencing data, all sample filtered reads were processed using the multiBamSummary function of deeptools 3.4.2 to calculate the scale factor for each sample. The scale factor and filtered reads for each sample were processed using the bamCoverage function of deeptools 3.4.2 to calculate the normalized coverage track. For single-cell sequencing data, the filtered reads were processed using the bamCoverage function of deeptools 3.4.2 to calculate the coverage track. The coverage tracks were processed using custom python3 scripts to calculate the coverage around the TSS sites.

## scRNA-seq data analysis

The human and mouse mixed reference file was from the 10x genomics website (https://cf.10xgenomics.com/supp/cell-exp/refdata-cellranger-hg19-and-mm10-3.0.0.tar.gz). The coding region of vector CEG was added to the reference file, and a new reference was made using the mkref function of cellranger 3.1.0. The human and mouse mixed reads were processed by cellranger 3.1.0 to obtain the UMI count matrix. The matrix files were processed by R package scater 1.16.2 using the function quickPerCellQC with the default set to filter low-quality cells. The cells were identified as human cells if the total UMI counts mapped to the mouse genome were less than 0.01 times the total UMI counts mapped to the human genome. The cells were identified as mouse cells if the total UMI counts mapped to the human genome were less than 0.018 times the total UMI counts mapped to

the mouse genome. The UMI counts were normalized by the library size. All of the following computations in this section were performed using MATLAB R2018b and associated toolboxes. Genes with mean counts under 1 CPM (counts per million) in any sample were neglected. Coefficients of variance (CV) and means for the remaining genes were computed for all three samples. The CV versus mean dot scatter plot was fitted to a power function ('power1') using the fit function. A cutoff line for identifying genes with CV outliers was defined as three σ above the fitted curve. The σ for each mean level was computed as the standard deviation of the CVs for the genes with a similar mean. The genes above this cutoff line (mean >5 CPM and CV >1.0), in any of the three samples, were filtered out for further analysis. The normalized count matrices from two control samples were pooled together as the control. If the CV of a gene was smaller than two in the control samples, it was grouped in the 'low CV' group. Otherwise, it was grouped in the 'high CV' group. Transcription factors (p300, CBP, and p65) for the target gene sets were identified from the ChIP-Atlas potential target genes with the option of ±1 kb distance from TSS. Genome assembly mm10 was selected as mouse reference in ChIP-Atlas. Genome assembly hg19 was chosen as a human reference in ChIP-Atlas. The UMI counts of filtered genes were processed by the R package saver 1.1.2 (*Huang et al., 2018*) to obtain a smooth distribution (*Figure 6G*).

## Statistical analyses

Data are described by the sample mean and standard deviation (SD). All statistical analyses were performed using the MATLAB 2018b statistics toolbox. Sample sizes and details of statistical tests are provided in the corresponding figure legends.

## Acknowledgements

This work was financially supported by the National Key R&D plan project (2019YFA0906002), by National Natural Science Foundation of China projects (31971181, 31770928), by the Shenzhen Peacock Plan (KQTD2016053117035204), by Science and Technology Planning Project of Guangdong Province, China (2016A050503010), and by Shenzhen Science and Technology Innovation Commission (ZDSYS20200811144002008). We thank Professors Qing Nie, Chunhui Hou, Wei Chen, and Liang Fang for stimulating discussions and suggestions, Messrs Wang Yuhao, Yi Zhang, Jiashun Xiao, and other members of the Huang group for technical assistance, and Miss Yifan Zhu for help with graphic illustration. We thank Professors Yi Yang, Jun Chu, and Jiandong Huang, and Jianbo Yue for providing plasmids for LightOn, mRuby3, and mCardinal tetR negative feedback plasmids, and mouse ES D3 cells. We acknowledge the technical support of Dr Wenjie Wei from the SUSTech Research Core Facility.

## Additional information

### Funding

| Funder | Grant reference number | Author |
| --- | --- | --- |
| National Key R&D plan project of China | 2019YFA0906002 | Wei Huang |
| National Natural Science Foundation of China | 31971181 | Wei Huang |
| National Natural Science Foundation of China | 31770928 | Wei Huang |
| Science and Technology Planning Project of Guangdong Province | 2016A050503010 | Wei Huang |
| Shenzhen Peacock Plan | KQTD2016053117035204 | Wei Huang |
| Shenzhen Science and Technology Innovation Commission | ZDSYS20200811144002008 | Xi Chen |

The funders had no role in study design, data collection and interpretation, or the decision to submit the work for publication.

## Author contributions
Deng Tan, Validation, Investigation, Methodology, Writing - review and editing; Rui Chen, Data curation, Software, Formal analysis, Methodology, Writing - review and editing; Yuejian Mo, Resources, Data curation, Software, Validation, Investigation, Visualization, Methodology, Writing - review and editing; Shu Gu, Validation, Investigation; Jiao Ma, Investigation; Wei Xu, Investigation, Writing - review and editing; Xibin Lu, Resources, Formal analysis, Methodology; Huiyu He, Formal analysis, Investigation; Fan Jiang, Resources, Software; Weimin Fan, Investigation, Methodology; Yili Wang, Resources, Software, Formal analysis, Methodology; Xi Chen, Resources, Software, Formal analysis, Investigation, Methodology, Writing - review and editing; Wei Huang, Conceptualization, Formal analysis, Supervision, Funding acquisition, Investigation, Methodology, Writing - original draft, Project administration, Writing - review and editing

## Author ORCIDs
Xi Chen https://orcid.org/0000-0003-2648-3146
Wei Huang https://orcid.org/0000-0002-6755-5807

## Decision letter and Author response
Decision letter https://doi.org/10.7554/eLife.65654.sa1
Author response https://doi.org/10.7554/eLife.65654.sa2

# Additional files
## Supplementary files
• Transparent reporting form

## Data availability
Sequencing data has been deposited in the European Nucleotide Archive under the accession-number PRJEB40269 and PRJEB41743. FACS source files for Figure 6 A-B have been deposited on Dryad with the DOI, 10.5061/dryad.z34tmpgf3. All data generated or analysed during this study are included in the manuscript and supporting files.

The following datasets were generated:

| Author(s) | Year | Dataset title | Dataset URL | Database and Identifier |
|---|---|---|---|---|
| Tan D, Chen R, Mo Y, Gu S, Ma J, Xu W, Lu X, He H, Jiang F, Fan W, Wang Y, Chen X, Huang W | 2021 | Noise modulation of mammalian inducible gene expression | https://www.ebi.ac.uk/ena/browser/view/PRJEB40269 | ENA, PRJEB40269 |
| Tan D, Chen R, Mo Y, Gu S, Ma J, Xu W, Lu X, He H, Jiang F, Fan W, Wang Y, Chen X, Huang W | 2021 | scRNA-seq for HeLa-AB1 and F9-AB2 cells | https://www.ebi.ac.uk/ena/browser/view/PRJEB41743 | ENA, PRJEB41743 |
| Tan D, Chen R, Mo Y, Gu S, Ma J, Xu W, Lu X, He H, Jiang F, Fan W, Wang Y, Chen X, Huang W | 2021 | FACS source files for Figure 6 A-B | https://doi.org/10.5061/dryad.z34tmpgf3 | Dryad Digital Repository, 10.5061/dryad.z34tmpgf3 |

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

## Appendix 1

### Construction of a 24-well quantitative blue-light illumination device

To enable dynamic and quantitative modulation of blue-light induction with minimized cross-sample interference, we constructed a 24-channel blue LED illumination apparatus for the black-wall transparent bottom 24-well plate (24-well μ-Plate: Catalog no. 82406; ibidi) (*Appendix 1—figure 1*). The 1-watt blue CREE LED (465 nm) and matching acrylic lens were from Zhongshan Jieneng LED lighting (Zhongshan, China). The current-intensity curve for each LED was calibrated with a SANWA laser power meter LP1, then fitted with a smoothing spline curve function (caps) using the curve-fitting toolbox in MATLAB 2018b, and saved as a lookup curve (*Appendix 1—figure 2*). The metal frame was computer numerical control (CNC)-machined from aluminum alloy, sandblasted, and darkened. It was assembled with screws and springs to allow tight fitting to the μ-Plate, which prevents cross-contamination of light between wells and easy assembly (*Appendix 1—figure 1*). A water-cooling system was used to maintain cell culture temperature at 37°C, independent of light intensity. A computer programmable 24-channel constant currents source (0–10 mA in 4096 steps) provided precise and independent control of currents for each LED. The control code was written in Python3 and modulated the currents at 1 Hz or slower. The light intensities at 'dark' were adjusted at the electric board to maintain them within 0.2 μW/cm$^2$.

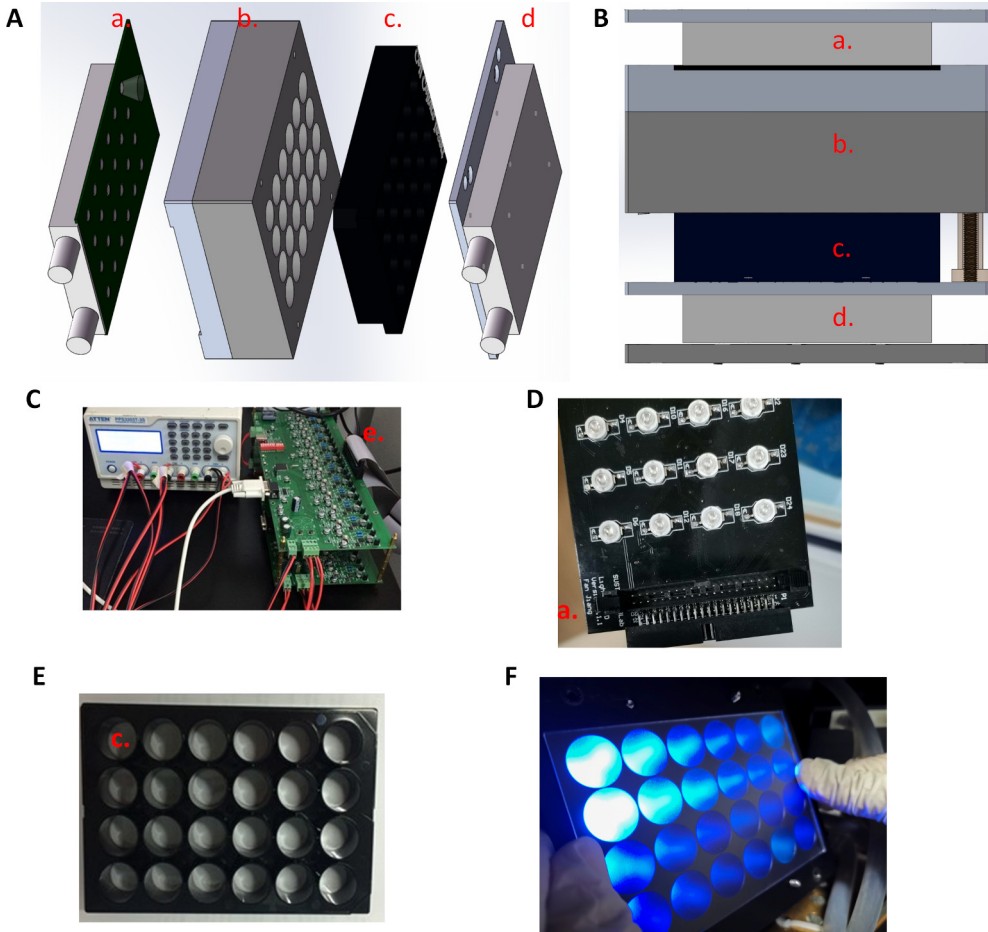

**Appendix 1—figure 1.** The design and construction of a 24-channel light illumination apparatus. (**A**) The assembly includes (a) a 24-channel blue LED board with lens and top water-cooling plate, (b) a matching 24-channel light guide plate to prevent cross-interferences between wells, (c) the ibidi 24-well μ-plate, and (d) a bottom water-cooling plate. (**B**) View of a fully assembled illumination apparatus. (**C**) The 24-channel programable constant current board and connected power supply.

*Appendix 1—figure 1 continued on next page*

*Appendix 1—figure 1 continued*

(**D**) The 24-channel blue LED board. (**E**) The ibidi 24-well μ-plate. (**F**) The 24-channel blue lights were visualized with a frosted glass plate.

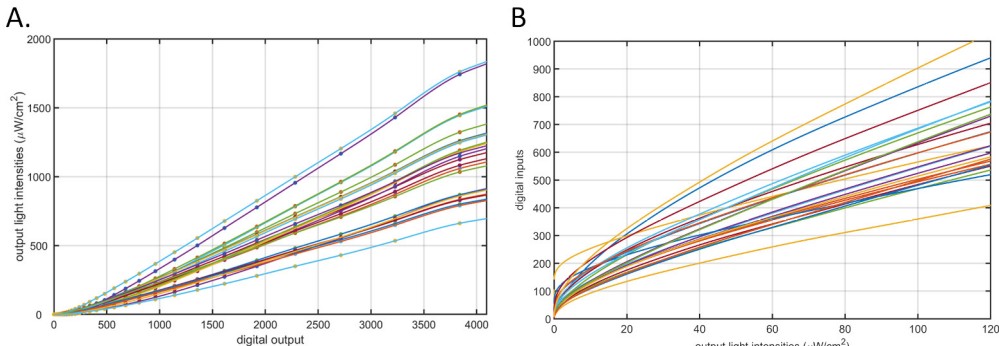

**Appendix 1—figure 2.** Calibration of the 24-channel light illumination apparatus. (**A**) Light intensity – digital input curves for the 24-channel LED board. The circles represent measurements made using a laser power meter. The lines are cubic smooth spline fitted curves. Each line represents a channel. (**B**) The lookup curves used to find digital input values (y-axis) for target light intensities (x-axis) for any of the 24 LEDs. Each line represents a channel.

## Construction of a 6-well quantitative blue-light illumination device for live-cell microscopy

We built a 6-channel LED illumination apparatus similar to the 24-channel ones used in the incubator, except that each LED was equipped with a 465/40 bandpass filter (Omega) and assembled on the cover of the environment chamber for multi-well plates (*Appendix 1—figure 3*). Matching blackout cloths with six holes of 16 mm in diameter, corresponding to six illuminating wells, were inserted between the cover of the 24-well μ-plate and the glass window of the cover for the environmental chamber to minimize cross-well light contamination further.

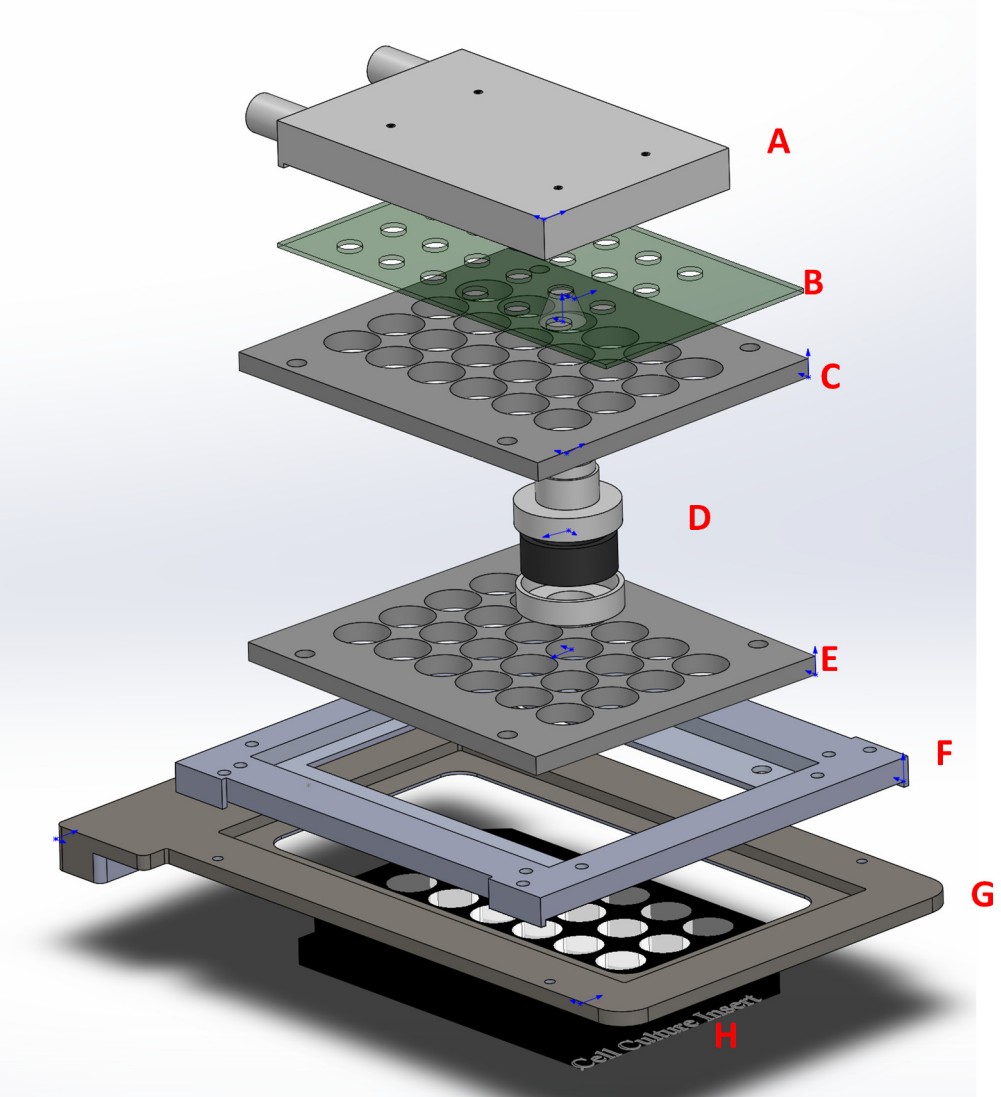

**Appendix 1—figure 3.** The six-channel light illumination apparatus used for live-cell imaging. The entire assembly consists of a water-cooling block (**A**), a 6-channel blue LED board with projection lens (**B**), matching 24-channel light guide plates (**C, E**), six bandpass filters (465/40) with holders (**D**), an adapter (**F**) for the cover of the environment chamber (**G**), and an ibidi 24-well µ-plate (**H**).

## Appendix 2

MATLAB code for computing predicted mRuby from single 'propagation' functions, including an empirical function fitted to experimental data and a hypothetical Hill function with high cooperativity.

```
figure();

% load flow cytometry data of Figure 1G, const light of 25 uW/cm2

% extracted from data files for Figure 1

Figure 1G = importdata('Figure 1G.csv');

G=Figure 1G.data(:,1);

R=Figure 1G.data(:,2);

% group cell into 20 groups with incremented GFP

[temp,j2]=sort(G);

G = G(j2);

R = R(j2);

N = length(j2); nd = floor(N/20); % numbers of cells per group

% calculate mean of each groups

for i2 = 1:20

 G1 = G((i2-1)*nd+(1:nd));

 R1 = R((i2-1)*nd+(1:nd));

 mG1(i2) = mean(G1);

 mR1(i2) = mean((R1));

end

subplot(3,3,1);

% experimental flow cytometry data

ksdensity([log10(G) log10(R)]);

%xlabel('log10(GFP-GAVPO)');

ylabel('log10(mRuby)');

view(2)

axis([4.8 6.3 2.4 5.4]);

title('experimental');

subplot (3,3,4);hold off

% experimental 'porpagation function' from GAVPO to mRuby

plot(log10(mG1),log10(mR1),'ro'); hold on

% fitted experimental data with mean mRuby vs mean GFP to a Hill function

% R = a*G^n/(G^n + k^n)+b;

a = 8.137e8; k = 1.865e8;b = 0;n = 1.856;

 x0 = 0e5:1e4:1.2e6;

 y0 = b + (a*x0.^n)./(x0.^n + k^n);

 plot(log10(x0),log10(y0),'r--');hold on;
```

```matlab
axis([5 6.2 2.8 5])
ylabel('log10(mRuby)');
legend(' GFP-mRuby',' Fitted curve','Location','northwest');
title('empirial propagation function');
subplot (3,3,7);hold off
% experimental 'transfer function' from GAVPO to mRuby
% hypothetical propagation function:
% Hill function with high cooperativity
a1 = 6.9e4; k1 = 6e5;b1 = 1e3;n1 = 8;
y1 = b1+(a1*x0.^n1)./(x0.^n1 +k1^n1);
plot(log10(x0),log10(y1),'k--');
axis([5 6.2 2.8 5])
xlabel('log10(GFP-GAVPO)');
ylabel('log10(mRuby)');
axis([5 6.2 2.8 5])
legend({['high' 10 'copperativity']},'Location','northwest');
title('hypothetical propagation function');
% calculate mRuby histograms from 'propagation function'
x = 1:0.1:8;
% for experimental distribution
hR = hist(log10(R),x)/length(G);
subplot (3,3,2);
plot(x,hR,'r-');
axis([2.4 5.4 0 0.1])
ylabel('freq.');
title('experimental histgram');
% for empirial propagation function
% mapping from GFP to mRuby
Rfit0 = b + (a*G.^n)./(G.^n + k^n);
% add lognormal noises with amplitudes proportional to means
Rfit0 = Rfit0.*(10.^(randn(N,1)*0.2));
% compute normalized histograms
hRfit0 = hist(log10(Rfit0),x)/length(G);
subplot (3,3,5);
plot(x,hRfit0,'r--');
axis([2.4 5.4 0 0.10])
ylabel('freq.');
title({['histgram predicted' 10 'by empirial propagation function']});
% for hypothetical propagation function:
```

```matlab
% Hill function with high cooperativity
% mapping from GFP to mRuby
 Rfit1 = b1+(a1*G.^n1)./(G.^n1 +k1^n1);
% add lognormal noises with amplitudes proportional to means
 Rfit1 = Rfit1.*(10.^(randn(N,1)*0.2));
% compute normalized histograms
 hRfit1 = hist(log10(Rfit1),x)/length(G);
 subplot (3,3,8);
 plot(x,hRfit1,'k--');
 axis([2.4 5.4 0 0.10])
 xlabel('log10(mRuby)');
 ylabel('freq.');
 title({['histgram predicted' 10 'by hypothetical propagation function']});
% histogram for 20 subpopulations of cells with narrow GFP ranges
% for experimental data, empirical and hypothetical propagation functions
x2 = 1:0.2:8;
for i2 = 1:20
 R1 = R((i2-1)*nd+(1:nd));
 % experimental data hR2(i2,:)=hist(log10(R1),x2)/nd;
 % prediction from empirical function
 Rfit0_1 = Rfit0((i2-1)*nd+(1:nd)); hRfit0_2(i2,:)=hist(log10(Rfit0_1),x2)/nd;
 % prediction from hypothetical function
 Rfit1_1 = Rfit1((i2-1)*nd+(1:nd));
 hRfit1_2(i2,:)=hist(log10(Rfit1_1),x2)/nd;
end
% experimental data
subplot (3,3,3)
plot(x2',hR2([1:2:19 20],:)+repmat(0.15*[0:10]',1,36));
axis([2 5.5 0 1.7]);
ax = gca; ax.YTick=[]
title({['experimental histgrams' 10 'from experiment'.10 'for GFP subpopulations']});
% prediction from empirical function
subplot (3,3,6)
plot(x2',hRfit0_2([1:2:19 20],:)+repmat(0.30*[0:10]',1,36));
axis([2 5.5 0 3.5])
ax = gca;
ax.YTick=[]
title({['histgrams' 10 'by empirial propagation function'.10 'for GFP subpopulations']});
% prediction from hypothetical function
```

```
subplot (3,3,9)
plot(x2',hRfit1_2([1:2:19 20],:)+repmat(0.36*[0:10]',1,36));
axis([2 5.5 0 4.1])
ax = gca;
ax.YTick=[];
xlabel('log10(mRuby)');
title({['histgrams' 10 'by hypothetical propagation function'.10 'for GFP subpopulations']});
```

# Appendix 3

**Appendix 3—key resources table**

| Reagent type (species) or resource | Designation | Source or reference | Identifiers | Additional information |
|---|---|---|---|---|
| Cell line (*Homo-sapiens*) | HeLa | Type Culture Collection of the Chinese Academy of Sciences, Shanghai | ATCC Cat# CCL-2; RRID:CVCL_0030 | |
| Cell line (*Mus musculus*) | F9 | Type Culture Collection of the Chinese Academy of Sciences, Shanghai | ATCC Cat# CRL-1720; RRID:CVCL_0259 | |
| Cell line (*M. musculus*) | D3 | American Type Culture Collection | ATCC Cat# CRL-1934; RRID:CVCL_4378 | |
| Cell line (*Homo-sapiens*) | HeLa-AB1 | This paper* | | See 'Construction of cell lines' in 'Materials and methods' |
| Cell line (*H.-sapiens*) | HeLa-AB2 | This paper* | | See 'Construction of cell lines' in 'Materials and methods' |
| Cell line (*H.-sapiens*) | HeLa-ABC2 | This paper* | | See 'Construction of cell lines' in 'Materials and methods' |
| Cell line (*H.-sapiens*) | HeLa-A1B1B3 | This paper* | | See 'Construction of cell lines' in 'Materials and methods' |
| Cell line (*H.-sapiens*) | HeLa-A1B1B4 | This paper* | | See 'Construction of cell lines' in 'Materials and methods' |
| Cell line (*H.-sapiens*) | HeLa-A3B2 | This paper* | | See 'Construction of cell lines' in 'Materials and methods' |
| Cell line (*H.-sapiens*) | HeLa-A4B5 | This paper* | | See 'Construction of cell lines' in 'Materials and methods' |
| Cell line (*H.-sapiens*) | HeLa-A4B6 | This paper* | | See 'onstruction of cell lines' in 'Materials and methods' |
| Cell line (*M. musculus*) | F9-AB2 | This paper* | | See 'onstruction of cell lines' in 'Materials and methods' |
| Cell line (*M. musculus*) | ES-D3 Rex1-GFP + Oct4-mCherry | This paper* | | See 'Evidence of dual-functional CBP/p300-induced heterogeneity in the regulation of endogenous genes' in 'Results' |
| Antibody | Anti-H3K9AC (rabbit, monoclonal) | Cell Signaling Technology | Cat# 9649; RRID:AB_823528 | ChIP-seq (1:50) |
| Antibody | Anti-H3K27me3 (rabbit, monoclonal) | Cell Signaling Technology | Cat#9733; RRID:AB_2616029 | ChIP-seq (1:50) |
| Antibody | Anti-H3K27AC (rabbit, polyclonal) | Abcam | Cat#ab4729; RRID:AB_2118291 | ChIP-seq (1:200) |
| Recombinant DNA reagent | GAVPO and 5xUAS promoter (plasmid) | *Wang et al., 2012* | | |

*Continued on next page*

*Appendix 3—key resources table continued*

| Reagent type (species) or resource | Designation | Source or reference | Identifiers | Additional information |
|---|---|---|---|---|
| Recombinant DNA reagent | mRuby3 (plasmid) | *Bajar et al., 2016* | | |
| Recombinant DNA reagent | mCardinal (plasmid) | *Chu et al., 2014* | | |
| Recombinant DNA reagent | pBX-CMVO2-TetRm CherryNLS (plasmid) | | Prof. Jiandong Huang, HKU | See 'Plasmid construction' in 'Materials and methods' and *Figure 1—figure supplement 1* |
| Recombinant DNA reagent | pX330 (plasmid) | Addgene | Cat# 42230; RRID:Addgene_ 42230 | |
| Recombinant DNA reagent | pBX-090 (plasmid) | *Lu and Huang, 2014* | | pGK-piggyBac-transposase |
| Recombinant DNA reagent | pBX-083 (plasmid) | *Lu and Huang, 2014* | | PB5-EF1a-GFP-nuc-PB3 |
| Recombinant DNA reagent | pBX-097 (plasmid) | *Lu and Huang, 2014* | | PB5-CAG-tdTomato-nuc-PB3 |
| Recombinant DNA reagent | pBX-123 (plasmid) | *Lu and Huang, 2014* | | PB5-SV40-puro-2A-rtTA-HS4-pTet-EGFPnuc-PB3 |
| Recombinant DNA reagent | Plasmid A1 (plasmid) | This paper* | | See 'Plasmid construction' in 'Materials and methods' and *Figure 1—figure supplement 1* |
| Recombinant DNA reagent | Plasmid A2 (plasmid) | This paper* | | See 'Plasmid construction' in 'Materials and methods' and *Figure 1—figure supplement 1* |
| Recombinant DNA reagent | Plasmid A3 (plasmid) | This paper* | | See 'Plasmid construction' in 'Materials and methods' and *Figure 1—figure supplement 1* |
| Recombinant DNA reagent | Plasmid A4 (plasmid) | This paper* | | See 'Plasmid construction' in 'Materials and methods' and *Figure 1—figure supplement 1* |
| Recombinant DNA reagent | Plasmid B1 (plasmid) | This paper* | | See 'Plasmid constructions' in 'Materials and methods' and *Figure 1—figure supplement 1* |
| Recombinant DNA reagent | Plasmid B2 (plasmid) | This paper* | | See 'Plasmid construction' in 'Materials and methods' and *Figure 1—figure supplement 1* |
| Recombinant DNA reagent | Plasmid B3 (plasmid) | This paper* | | See 'Plasmid constructions' in 'Materials and methods' and *Figure 1—figure supplement 1* |
| Recombinant DNA reagent | Plasmid B4 (plasmid) | This paper* | | See 'Plasmid construction' in 'Materials and methods' and *Figure 1—figure supplement 1* |
| Recombinant DNA reagent | Plasmid B5 (plasmid) | This paper* | | See 'Plasmid construction' in 'Materials and methods' and *Figure 1—figure supplement 1* |

*Continued on next page*

*Appendix 3—key resources table continued*

| Reagent type (species) or resource | Designation | Source or reference | Identifiers | Additional information |
|---|---|---|---|---|
| Recombinant DNA reagent | Plasmid B6 (plasmid) | This paper* | | See 'Plasmid construction' in 'Materials and methods' and *Figure 1—figure supplement 1* |
| Recombinant DNA reagent | Plasmid C2 (plasmid) | This paper* | | See 'Plasmid construction' in 'Materials and methods' and *Figure 1—figure supplement 1* |
| Recombinant DNA reagent | Plasmid D1 (plasmid) | This paper* | | See 'Plasmid construction' in 'Materials and methods' and *Figure 1—figure supplement 1* |
| Recombinant DNA reagent | Plasmid D2 (plasmid) | This paper* | | See 'Plasmid construction' in 'Materials and methods' and *Figure 1—figure supplement 1* |
| Recombinant DNA reagent | gRNA-Rex1-Cas9 (plasmid) | This paper* | Cas9 and gRNA for targeting Rex1 | See 'Plasmid construction' in 'Materials and methods' |
| Recombinant DNA reagent | gRNA-Oct4-Cas9 (plasmid) | This paper* | Cas9 and gRNA for targeting Oct4 | See 'Plasmid construction' in 'Materials and methods' |
| Commercial assay or kit | Sphero Rainbow calibration beads six peaks | Spherotech | Cat#RCP-60–5 | |
| Commercial assay or kit | NEXTFLEX ChIP-Seq Library Prep Kit for Illumina Sequencing | PerkinElmer | Cat#NOVA-514120 | |
| Commercial assay or kit | TB Green Premix Ex Taq II | Takara | Cat#RR820B | |
| Commercial assay or kit | PureLinkTM Genomic DNA Mini Kit | Invitrogen | Cat#K1820-01 | |
| Commercial assay or kit | TTE Mix V50 | Vazyme | Cat#TD501 | |
| Commercial assay or kit | Nextera kit | Illumina | Cat#FC-121–1030 | |
| Commercial assay or kit | Chromium Next GEM Single Cell 3' GEM, Library and Gel Bead Kit v3.1 | 10X Genomics | Cat#PN-1000128 | |
| Commercial assay or kit | Chromium Next GEM Chip G Single Cell Kit | 10X Genomics | Cat# PN-1000127 | |
| Commercial assay or kit | Single Index Kit T Set A | 10X Genomics | Cat#PN-1000213 | |
| Chemical compound, drug | Zeocin | Gibco | Cat#R25001 | |
| Chemical compound, drug | Doxycycline | Sigma | Cat#D9891 | |
| Chemical compound, drug | Puromycin | Gibco | Cat#A1113803 | |
| Chemical compound, drug | LMK-235 | SelleckChem | Cat#S7569 | |
| Chemical compound, drug | Vorinostat | SelleckChem | Cat#S1047 | |

*Continued on next page*

*Appendix 3—key resources table continued*

| Reagent type (species) or resource | Designation | Source or reference | Identifiers | Additional information |
|---|---|---|---|---|
| Chemical compound, drug | PF-06726304 | SelleckChem | Cat#S8494 | |
| Chemical compound, drug | A-485 | SelleckChem | Cat#S8740 | |
| Chemical compound, drug | CHIR99021 | Stemolecule | Cat#04–0004 | |
| Chemical compound, drug | PD0325901 | Stemolecule | Cat#04–0006 | |
| Chemical compound, drug | DMSO | Sigma | Cat# D8418 | |
| Chemical compound, drug | Digitonin | Promega | Cat#G9441 | |
| Software, algorithm | MATLAB | Mathworks | Matlab R2018b; RRID:SCR_001622 | |
| Software, algorithm | FlowJo | FlowJo | FlowJo X; RRID:SCR_008520 | |
| Software, algorithm | Scripts for sequencing data analysis | This paper* | | https://github.com/QBioLab/sequence-data-analysis-for-noise-control (copy archived at swh:1:rev:c20430824c5a603eec695ee167e32e4cd1b6c024), *Jiang, 2020a* |
| Software, algorithm | Live cell mRNA image processing codes | This paper* | | https://github.com/QBioLab/CountmRNA.jl, (copy archived at swh:1:rev:27fa41066bf09a3c1ca00fd333f72755e481b16f); *Jiang, 2020b* |
| Software, algorithm | Live cell tracking and single-cell fluorescent protein quantification | This paper* | | https://github.com/QBioLab/FindYourCell.jl, (copy archived at swh:1:rev:0fee217824a5d6c2f12871477c514ef55b744b09); *Jiang, 2021* |
| Software, algorithm | Simulation and analysis codes for the ODE model | This paper* | | https://github.com/QBioLab/noise-simulation, (copy archived at swh:1:rev:46301588f603073c2982a5e912afa4a3274bf207); *Jiang, 2020c* |
| Other | DMEM | Corning | Cat#10–013-CV | |
| Other | Fetal bovine serum (FBS) | Gibco | Cat#10270–106 | |
| Other | ES cell FBS, US origin | Gibco | Cat#16141–079 | |
| Other | Leukemia inhibitory factor (LIF) | Millipore | Cat#ESG1107 | |
| Other | GlutaMax-I | Gibco | Cat#A12860 | |
| Other | MEM non-essential amino acid (NEAA) | Gibco | Cat#1140–050 | |
| Other | 2-Mercaptoethanol | Gibco | Cat#31350–010 | |

*Continued on next page*

*Appendix 3—key resources table continued*

| Reagent type (species) or resource | Designation | Source or reference | Identifiers | Additional information |
|---|---|---|---|---|
| Other | TrypLE Express Enzyme | Gibco | Cat#12604–021 | |
| Other | Lipofectamine 3000 | Invitrogen | Cat#L3000-015 | |
| Other | Penicillin/Streptomycin | Gibco | Cat#15140–122 | |
| Other | μ-plate 24-well plate | ibidi | Cat#82406 | |
| Other | H3K27AC Chip-seq, ATAC-seq and sc-ATAC-seq data | This paper* | | European Nucleotide Archive under the accession-number PRJEB40269 |
| Other | scRNA-seq rawdata for HeLa-AB1 and F9-AB2 cells | This paper* | | European Nucleotide Archive under the accession-number PRJEB41743 |

*These reagents or resources in this paper can be obtained by contacting the corresponding author Dr Wei Huang (huangw@sustech.edu.cn).

