## [Decision Letter]

**Acceptance summary:**

This paper uses a light-induced, synthetic gene expression system in mammalian cells to show that mean gene expression and variability ('noise') of expression can be independently tuned by modulating the light input. This expands this general strategy from yeast to mammalian cells and provides a tool to study the functional consequences of expression variability in mammalian cells. The paper also reports an impressive amount of single-cell data on gene expression and chromatin state which suggest that variations in histone acetylation state contribute to the expression variability.

**Decision letter after peer review:**

Thank you for submitting your article "Quantitative Control of Noise in Mammalian Gene Expression by Dynamic Histone Regulations" for consideration by *eLife*. Your article has been reviewed by 3 peer reviewers, and the evaluation has been overseen by a Reviewing Editor and Kevin Struhl as the Senior Editor. The reviewers have opted to remain anonymous.

This paper offers two main results:

(1) proposed model of bistability via CBP/p300 positive feedback

(2) reducing bimodality via PWM of light.

Result #2 is new for the mammalian system but has previously been shown in yeast. Result #1 requires additional data to be shown conclusively.

Essential revisions:

1) The proposed model has weaknesses that need to be addressed: (i) the model is conceptually not clear enough, (ii) the corresponding mathematical model is not calibrated with experimental data, (iii) some of the existing data seem to contradict the model, (iv) the model needs to be more thoroughly tested.

(i) Conceptual consistency and clarity should be improved: in a deterministic setting, bistability in positive feedback loops is well-defined, but the effect of PWM would be a stabilization of an unstable steady-state, as in https://www.nature.com/articles/s41467-017-01498-0. In the stochastic setting, stochastic promoter switching would be sufficient, and bistability is ill-defined, as demonstrated in https://www.nature.com/articles/s41467-018-05882-2#Sec25. Positive feedback via histone modifications could be consistent with either scenario, and depends on the experimental evidence for alternate, deterministically stable steady states. The experimental data in Figures 1D and 3D only weakly support the notion of 'low' and 'high' alternate states (especially because protein abundances are bounded from below and potentially also from above due to general expression capacity or simply applied input ranges), compared to the simulated data in Figure 2F. More direct evidence for bistability would be given by experiments that demonstrate hysteresis, which should be considered-see (iv).

(ii) The ODE model presented is not calibrated with experimental data and therefore of limited value for data interpretation. The authors should either (1) de-emphasize the mathematical model, acknowledging its limitations, or (2) include a calibrated mathematical model-for example, using a stochastic model as in Benzinger and Khammash. The inclusion of multiple reporters (see iii) would be a benefit. The authors could fit to their flow/genomic data to obtain further insights into how/why PWM of light suppresses noise. The current model represents cell-to-cell variability by cell-specific parametrization, and hence only contains extrinsic noise components.

(iii) Multiple copies of GAVPO-mRuby reporter in HeLa-AB1 strain and the measured bimodal response suggest that the current model is wrong. According to the authors' proposed model, each integrated GAVPO reporter gene (mRuby) should be independently bistable because local GAVPO binding and chromatin modifications regulate mRuby expression in cis. Figure 3—figure supplement 1 shows that there are 9 copies of GAVPO-mRuby plasmid B1 integrated at different locations in HeLa-AB1 strain. Thus, at intermediate AM light intensity, the mRuby gene expression should be deca-modal (10 modes), not bimodal (as shown in Figures 1 – 2). Something seems inconsistent between the local chromatin positive feedback model and the observed data.

The mathematical model used to validate the observations does not model the total expression from 9 independent promoters, which is a critical omission given the cis-nature of the positive feedback loop. The fact that these 9 promoters generate 2 peaks at intermediate light intensity suggests that the GAVPO bistability likely originates from a trans-effect, i.e., either all 9 promoters are OFF or all 9 promoters are ON, not a cis-effect.

In addition, from Figure 1—figure supplement 2G, it appears that when cells are exposed to light, it is mostly the cells containing a higher amount of GFP-GAVPO that switch into the mRuby ON state. Is it possible that the bistability simply comes from the dimerization event caused by light?

(iv) A bimodal response is consistent with bistability, but it does not prove bistability. To unambiguously show bistability, the authors should (1) measure and show dynamic signatures of bistability (e.g., hysteresis and/or following single cell dynamics to prove signature of two stable basins of attraction with transitions at some threshold GAVPO concentration) and (2) test their proposed model with precise, mechanistic mutations. The most compelling evidence in favor of their model is the use of the A-485 inhibitor of HAT activity of p300/CBP (i.e., prevent positive feedback loop) to convert the bimodal profile into a unimodal and graded profile. However, this is a crude perturbation because A-485 likely affects the expression of many genes, including any cis- and trans-factors that affect mRuby gene expression. Mutations specific to GAVPO and mRuby promoter that destroy positive feedback would be the cleanest perturbation.

Suggestions:

1. Replace the p65A activation domain with mutants that activate transcription by recruiting RNA Pol II but which cannot recruit CBP/p300 and modify histones, e.g. "Minimal activators that bind to the KIX domain of p300/CBP identified by phage display screening", Frangioni et al., Nature Biotechnology (2000). If such p65A mutants don't exist, there are VP16 activation domains that can separate p300/CBP-recruiting versus TFII-recruiting functions, e.g. "The H1 and H2 regions of the activation domain of herpes simplex virion protein 16 stimulate transcription through distinct molecular mechanisms, Ikeda et al., Genes to Cells (2002). Prediction: If p300/CBP recruitment is essential for positive feedback, then you will no longer see a bistable response. If you see bistability, then an alternative should be considered more seriously.

2. Use a monomeric transcription factor regulated by a chemical (e.g. Zinc-finger-ER-p65) and engineer a mRuby promoter with a single binding site. Prediction: This system should exhibit bistable mRuby response at intermediate hormone concentrations, if your model is correct. NOTE: The authors already use an rtTA doxycycline-inducible system in Figure 2—figure supplement 2, which has VP16 activation domain, and exhibits a bimodal distribution at intermediate doxycycline concentrations. The authors could modify VP16 to abolish p300/CBP recruitment of the rtTA construct.

2) In its current form, Figure 5 does not convincingly show that PWM reduces noise:

(i) In Panel E, at time 0 (no light induction), AM 100 uW, 25 uW and PWM start at different CV values. It is unclear why this is the case especially since this is not observed in the mRuby CV. This needs to be addressed.

(ii) The authors should add a panel to Figure 5 showing the mean mRNA levels for the three respective cases, AM 100 uW, 25 uW and PWM 200m.

(iii) A discussion should be included about the large number of cells showing almost no mRNA for AM 25 uW and PWM 200m, does this means that there are cells trapped in the off state, and how do these cells impact the CV calculations?

(iv) Since the pulsatile behavior appears in ~15% of the cells (Figure 5C), how does the CV time course behave if CV is only calculated for these cells exhibiting pulsatile behavior.

3) Figure 1E/F and around line 120: The argument of 'limited noise of GFP-GAVPO' requires more details, namely a quantitative analysis of this noise and the effects of noise propagation – the current analysis focuses on relations between means and it is not clear what the effect of GAVPO expression noise on mRuby noise is. In addition, TetR-GFP-nuc may not accurately reflect GAVPO noise if the stabilities of the two proteins are different.

4) TetR-noise reducing circuit (or Plasmid A1). The design or purpose of this plasmid is a little confusing. The TetR-GFP-nuc synthetic transcription factor binds to its CMV promoter to repress its own transcription (negative feedback loop). This should repress mean gene expression and "squeeze" the expression variance about the mean (ref: Becskei et al., Nature 2000), i.e. noise reducing. However, the authors added doxycycline one day before the start of light induction and then maintained it throughout the experiment. This would destroy the negative feedback loop and, thus, increase the TetR and GAVPO mean and variance back to unrepressed CMV promoter levels. Are the authors using this circuit to keep TetR and GAVPO low until the start of the experiment (conditional expression) or are the authors using this circuit to reduce variance in TetR and GAVPO levels (noise-reduction)? Please clarify how and why you are using this circuit.

5) Although not essential in a revised manuscript, testing systems beyond light-regulated GAVPO would strengthen the generality of the authors' conclusions. The good news is that their rtTA system (which has a VP16 trans-activation domain that recruits both p300/CBP and RNA Poll II) exhibits bimodal distribution at intermediate concentrations of doxycycline (Figure 2—figure supplement 2), so they have a different system, strains, and preliminary results to test the generality of their model.

Presentation:

6) The text requires thorough language editing throughout. Widespread grammatical mistakes-in particular on verb tense and singular/plural-make it difficult to follow the arguments.

7) If the authors decide to include a calibrated mathematical model, the model should be more clearly described in the main text (not only in the Supplementary Material).

8) A brief description of how the light inducible GAVPO system works would be helpful either in the introduction or the beginning of the Results section. Perhaps adding Figure 1—figure supplement 1 to Figure 1.

9) Please add a schematic in Figure 1 or the supplement that clearly illustrates how the light is pulsed (i.e., duration of on time and off time), the light intensity used per pulse, and the mean light intensity. Something similar to Figure 4 D-E, yet for all period lengths.

10) Please explain and/or motivate the use of different PWM regimes to provide evidence for different aspects of the work, e.g., line 137 400min period, line 261 600min, line 296, 200min).

11) Figure 2—figure supplement 1 is a confusing figure because it makes two different points. The first half (A-F) shows that LMK-235 has a minor impact on noise reduction. The second half (G-L) shows that a different strain F9-AB2 shows identical signatures to the HeLa-AB1 strain, i.e. bimodality at intermediate levels of AM light, which is reduced to unimodal by FM light. Perhaps split these data into separate supplementary figures? In lines 205 – 207, these data are referred to as "figure supplement 2".

12) In mouse embryonic stem cells (Figure 1—figure supplement 3), there is only a clear noise reduction effect with 200 and 400 minutes, yet for these there is also a decrease in mean GFP levels. Therefore, the ability to modulate noise independent of mean seems not to hold in mouse embryonic stem cells. Please state this more clearly (text lines 149-154).

13) Please provide more detailed explanations of the somewhat puzzling chromatin state dynamics in Figure 4 where a longer-term drift (denoted as epigenetic 'memory') is apparent. How do these dynamics relate to the data presented in Figure 1 (the Methods seem to suggest a measurement time at 48h, but this is not specified in the captions).

14) The title 'RNA dynamics flattens PWM-induced pulsatile chromatin opening' implies an influence of RNA dynamics on chromatin state, which is not plausible given the construction of the synthetic system (the mRNA does not encode a protein that could modify chromatin state) and the postulated model (positive feedback is established via TF binding, not mRNA expression). mRNA dynamics will impact protein dynamics, and the title and text should reflect this aspect.

15) Figure 2B and Line 943: If there are only 4 data points from 4 independent experiments, why not show them rather than summarize their statistics with a box-whisker plot?

16) Line 303/Figure 5C: The export of mRNA from the nucleus to the cytoplasm, which is mentioned in the text, is not shown in the figure that is cited.

17) Line 248: cite Figure 3H in addition to Figure 3—figure supplement 1A.

18) Please write out "min" instead of "m" for the pulse periods in the figure legends.

19) Prior work on the use of PWM and available explanations for its noise-reducing effect should be mentioned and discussed more explicitly.

20) A publication that came out recently and also describes independent control of mean expression and variability could be cited: https://doi.org/10.1038/s41467-020-20467-8.

[Editors' note: further revisions were suggested prior to acceptance, as described below.]

Thank you for resubmitting your work entitled "Quantitative Control of Noise in Mammalian Gene Expression by Dynamic Histone Regulations" for further consideration by *eLife*. Your revised article has been evaluated by all three original reviewers, Kevin Struhl as the Senior Editor, and a Reviewing Editor.

Thank you for performing additional experiments. The manuscript has been improved but remaining issues make the paper not acceptable for publication in *eLife* in its current form.

In particular, all reviewers felt that bistability of the system is still not well enough supported to make this a major claim of the paper.

On the other hand, your findings on PWM-mediated noise suppression in mammalian cells and the link to histone modifications are interesting and large enough an advance for publication. Your data strongly support the idea that CBP/p300 recruitment via chromatin acetylation leads to bimodal gene expression at intermediate GAVPO activity and that PWM of GAVPO activity via light can sculpt the bimodal output to be unimodal.

While you could perform additional experiments to attempt to demonstrate bistability (see under point 2 below), we recommend to remove this claim from the paper, replace "bistable" with "bimodal", remove the ODE model, and discuss your findings in a more balanced way that leaves room for alternative interpretations.

If you want to maintain your conclusion regarding bistability with the current data, you will need to submit to another journal as a home for your paper.

Necessary changes for a revision of your paper for *eLife* include:

– Re-writing the second half of the abstract, lines 23 – 29.

– Re-writing the end of the Introduction, lines 83 – 96.

– Considering the criticism of the data in Figure 1H (see below) and modify that section (line 123-137).

– Line 227 – 236: Make it clear that this is a working model, but other models are possible. (This may then be better suited for the discussion.)

– Line 237 – 273: Please see comments under point 2 below. These experiments do not strongly support bistability and the conclusions need to be re-phrased. The model (lines 284-295) could be removed.

– Modify the discussion in lines 493 – 507 and 518-539, since these conclusions are not well supported.

– Discussion paragraph starting in line 540 needs an introduction.

– More of the limitations in the experiments that you discuss in the response to the reviewers (e.g. point 2, 3, 10, 13 in the previous decision letter) could show up in the paper itself, so that readers are made aware of those.

– Generally, re-assess all mentions of bistability, hysteresis and positive feedback loop.

Note that, while we point out specific passages to modify, you may want to re-consider the structure of your paper given these major changes.

Specific criticisms:

(1) Figure 1H aims to demonstrate that bistability and not cooperativity in gene expression (as in Benzinger and Khammash) underlies the observed population distributions by propagating GAVPO distributions through empirical (from Figure 1G data) and hypothetical high-cooperativity transfer functions to predict mRuby distributions. Details on methods are missing (and should be provided), but clearly the mRuby monomodal distribution for the empirical transfer function does not match (approximately) the distribution for AM, 20\muW/cm^2^ in Figure 1F and it is unclear if it does so for the mRuby marginal in Figure 1G (the raw data for this panel is not available), as would be expected. Furthermore, it is unclear, how the predicted mRuby marginal for high cooperativity was obtained. Very approximate simulations for both scenarios (Matlab code below) rather indicate the opposite of the authors' conclusions when compared to data in Figure 1F,G.

Matlab code on point (1):

%% cooperativity simulations (x: GAVPO, y: mRuby)

ns = 1e4;

d0 = table2array(readtable('88763_1_data_set_2024261_qtg66v.xlsx','sheet','Fig1H-3','range','a2:b122'));

cdf = cumsum(d0(:,2));

[~,idx] = unique(cdf);

x = interp1(cdf(idx),d0(idx,1),rand(ns,1));

% loop: cooperativity

hill = {@(x,k,n) 3 + 1*(x-4.5);.…

@(x,k,n) 3 + 2*x.^n./(k.^n+x.^n)};

n = 70;

k = 5.7;

figure();

for z = 1:length(hill)

subplot(3,2,z)

xi = linspace(4.5,6.5,100);

plot(xi,hill{z}(xi,k,n));

subplot(3,2,z+2)

y = hill{z}(x,k,n) +.25*randn(size(x));

ksdensity([x,y]);

view(2);

axis([4.5 6.5 2 6])

subplot(3,2,z+4);

histogram(y,'Normalization','pdf');

hold on;

histogram(x,'Normalization','pdf');

end

(2) Figure 2 aims to demonstrate bistability and hysteresis as proposed by the ODE model via single-cell time lapse microscopy (2F-H) and FACS (2I,J). The single-cell trajectories in Figure 2G, however, do not represent the population distribution. They were selected according to 'low' or 'high' mRuby signal at 24h; the data only demonstrates the dynamics for reaching the target state, and not bistability. For experiments demonstrating hysteresis, in the regime of bistability, one would expect distinct (bimodal under noise) distributions close to the 'low' or 'high' starting state. This is clearly not the case given the data in Figure 2I,J. The slight shift to higher mRuby in the intermediary regime (10-25 light intensity) used as an argument for hysteresis in the manuscript can be easily explained by the population not being in steady-state because the distribution for the predicted monostable 0 light input case is right-shifted by prior stimulation as well.

In brief, the single-cell experiments (Figure 2F-H) show a bimodal induction response. The OFF cells stay OFF and the ON cells stay ON. It would have been more convincing to see an induced ON cell stochastically cross a threshold and then return to the OFF state. Or the authors could have started with a fully induced population (100% ON with 100 uW/cm^2^ induction) then put at intermediate light and filmed single cells. The expectation would be that a fraction of the cells crosses the threshold and goes to the stable OFF state. Alternatively, the authors could have sorted ON and OFF cells, and then measured the time evolution of mRuby/mCardinal distribution of the OFF population and ON population.

---

## [Author Response]

This paper offers two main results:(1) proposed model of bistability via CBP/p300 positive feedback(2) reducing bimodality via PWM of light.Result #2 is new for the mammalian system but has previously been shown in yeast. Result #1 requires additional data to be shown conclusively.

We agree that the PWM *per se* has been reported in yeast. We initially discovered the effect of long-period PWM on noise reduction in mammalian cells back in 2016, before the publications of yeast and *E. coli.* PWM studies. We believe that our phenomena and mechanisms are sufficiently different from the yeast study.

We were also puzzled by the potential mechanisms for a long time before finding the histone acetylation. Our experiment observed period (≥400 min) of PWM is much longer than the yeast study (30 min) and much shorter than the typical epigenetic memory consensus in the community. Our work identified CBP/p300 as the fast epigenetic modulator that contributes to gene expression heterogeneity.

In the yeast paper, the authors observed bimodality (Figure 3h in Benzinger and Khammash paper) only when they intentionally increase TF variability by introducing plasmid copy number variation. In this scenario, the cells are no longer isogenic.

Essential revisions:1) The proposed model has weaknesses that need to be addressed: (i) the model is conceptually not clear enough, (ii) the corresponding mathematical model is not calibrated with experimental data, (iii) some of the existing data seem to contradict the model, (iv) the model needs to be more thoroughly tested.(i) Conceptual consistency and clarity should be improved: in a deterministic setting, bistability in positive feedback loops is well-defined, but the effect of PWM would be a stabilization of an unstable steady-state, as in https://www.nature.com/articles/s41467-017-01498-0. In the stochastic setting, stochastic promoter switching would be sufficient, and bistability is ill-defined, as demonstrated in https://www.nature.com/articles/s41467-018-05882-2#Sec25. Positive feedback via histone modifications could be consistent with either scenario, and depends on the experimental evidence for alternate, deterministically stable steady states. The experimental data in Figures 1D and 3D only weakly support the notion of 'low' and 'high' alternate states (especially because protein abundances are bounded from below and potentially also from above due to general expression capacity or simply applied input ranges), compared to the simulated data in Figure 2F. More direct evidence for bistability would be given by experiments that demonstrate hysteresis, which should be considered-see (iv).

Thank you very much for the insights. We need to make a more explicit description of the concept of our model (Figure 2D). In addition to adding a separate paragraph in the Discussion section (Line 517-538 in the revised manuscript), we also performed additional experiments suggested in point (iv) to direct validate the properties of bistability, which we will describe later in this letter. The concept of stabilized unstable steady states was explicitly developed in control chaos in physical systems, exemplified with the famous OGY theory https://journals.aps.org/prl/abstract/10.1103/PhysRevLett.64.1196. Their motivation was to feedback control chaos with small perturbation in the systems. Non-feedback control with small periodic perturbation has also been shown to control chaos by stabilizing unstable steady state, as in https://journals.aps.org/prl/abstract/10.1103/PhysRevLett.72.96. The periodic control work in *E. coli* to stabilize an unstable steady state could be considered as an extension of these concepts in complex biological systems. In addition, the location of unstable steady state in a bistable system would decrease with increasing light intensity, as shown in the black dashed line in Figure 2E, inconsistent with our result that the mean mRuby is monotonically correlated with the mean light intensity (Figure 1C).

It is correct that the hypothesis proposed by Benzinger and Khammash in the yeast paper that the cooperative TF-gene expression curve is sufficient to generate bimodal distribution with high variable TF distribution, shown in Figure 1G. But it is not the case in our study. In our experiment, GAVPO and mRuby expression do not exhibit clear cooperativity.

(ii) The ODE model presented is not calibrated with experimental data and therefore of limited value for data interpretation. The authors should either (1) de-emphasize the mathematical model, acknowledging its limitations, or (2) include a calibrated mathematical model-for example, using a stochastic model as in Benzinger and Khammash. The inclusion of multiple reporters (see iii) would be a benefit. The authors could fit to their flow/genomic data to obtain further insights into how/why PWM of light suppresses noise. The current model represents cell-to-cell variability by cell-specific parametrization, and hence only contains extrinsic noise components.

Thanks for point out the limitations of our deterministic ODE model, including not able to simulate the accurate heterogeneous distributions with intrinsic noise. We also could simulate the nine insertions of 5xUAS-mRuby independently. Our main intention is to use it to find out if inhibition of HAT could reduce noise. It is validated with experiments. We have built a stochastic model containing all the processes in our ODE model, and have written the code to simulate using a stochastic simulation algorithm similar to the yeast paper, and performed a good deal of optimization. Unfortunately, it would be computationally prohibitive expensive for us to carry it out with current resources. To compare with the AM and 400 min PWM data, including the distributions and noises, we need to simulate 1000 cells for 48 hours. For one set of parameters, it took 100 CPU seconds to simulate the AM and 400 min PWM conditions. To ensure finding the “global minimum” in fitting to experimental data, using the parallel tempering Monte-Carlo method (https://pubmed.ncbi.nlm.nih.gov/19810318/), it would take in the vicinity of hundreds of thousands of node-hours.

(iii) Multiple copies of GAVPO-mRuby reporter in HeLa-AB1 strain and the measured bimodal response suggest that the current model is wrong. According to the authors' proposed model, each integrated GAVPO reporter gene (mRuby) should be independently bistable because local GAVPO binding and chromatin modifications regulate mRuby expression in cis. Figure 3—figure supplement 1 shows that there are 9 copies of GAVPO-mRuby plasmid B1 integrated at different locations in HeLa-AB1 strain. Thus, at intermediate AM light intensity, the mRuby gene expression should be deca-modal (10 modes), not bimodal (as shown in Figures 1 – 2). Something seems inconsistent between the local chromatin positive feedback model and the observed data.The mathematical model used to validate the observations does not model the total expression from 9 independent promoters, which is a critical omission given the cis-nature of the positive feedback loop. The fact that these 9 promoters generate 2 peaks at intermediate light intensity suggests that the GAVPO bistability likely originates from a trans-effect, i.e., either all 9 promoters are OFF or all 9 promoters are ON, not a cis-effect.In addition, from Figure —figure supplement 2G, it appears that when cells are exposed to light, it is mostly the cells containing a higher amount of GFP-GAVPO that switch into the mRuby ON state. Is it possible that the bistability simply comes from the dimerization event caused by light?

Thanks for point out this critical insight. According to our model, for the HeLa-AB1 cell with nine insertions of 5xUAS-mRuby, theoretically, we should expect 9 ”on “ peaks and 1 “low” peak in the mRuby distribution. We suspect that multiple “high” peaks were convoluted into a large expanded peak due to variable expression of the “high” state at each site. This convoluted “high” peak exhibit a gradual shift to the right with increasing light intensity, visible in Figure 1F, Figure 2I-J, and Figure 1—figure supplement 4A. We use a simple emulation to illustrate the effect, shown in Figure 3—figure supplement 2, assuming the expression levels and variability (CV=0.47) of each site are identical at the “high” states and ignore the stochastic transition between on and off states. If the CVs for each site were reduced to 0.35, we could start to observe multiple “high” peaks, shown in Author response image 1. In this entire study, the lowest CV with any reporter fluorescent protein is about 0.4. In addition, we generate three clones with ~3, ~5, and ~11 copies of 5xUAS-mCardinal insertions using the HeLa-AB1 cell line. They also exhibit similar mCardinal distributions under increase AM light inductions (Figure 1—figure supplement 4), which is consistent with our model. We add a separate paragraph in the discussion session to clear up this puzzle (Line 539-551 in the revised manuscript).

**Author response image 1. sa2fig1:** Emulation of combinations of ‘high’ states at multiple sites.

We calculated the mean GFP-mRuby response curves, there are fitted well to power functions with approximately mRuby~GFP^1.8^ and don't look like a sigmoid curve that reaches a plateau (i.e., shown in Figure 1G and Figure 2C). The log-log flow cytometry plots are misleading because they suppressed the difference in mRuby, make them look like they plateaued at low and high ends of GFP. The light intensities are apparently not close to inducing saturated dimerization in our experiments.

(iv) A bimodal response is consistent with bistability, but it does not prove bistability. To unambiguously show bistability, the authors should (1) measure and show dynamic signatures of bistability (e.g., hysteresis and/or following single cell dynamics to prove signature of two stable basins of attraction with transitions at some threshold GAVPO concentration) and (2) test their proposed model with precise, mechanistic mutations. The most compelling evidence in favor of their model is the use of the A-485 inhibitor of HAT activity of p300/CBP (i.e., prevent positive feedback loop) to convert the bimodal profile into a unimodal and graded profile. However, this is a crude perturbation because A-485 likely affects the expression of many genes, including any cis- and trans-factors that affect mRuby gene expression. Mutations specific to GAVPO and mRuby promoter that destroy positive feedback would be the cleanest perturbation.Suggestions:1. Replace the p65A activation domain with mutants that activate transcription by recruiting RNA Pol II but which cannot recruit CBP/p300 and modify histones, e.g. "Minimal activators that bind to the KIX domain of p300/CBP identified by phage display screening", Frangioni et al., Nature Biotechnology (2000). If such p65A mutants don't exist, there are VP16 activation domains that can separate p300/CBP-recruiting versus TFII-recruiting functions, e.g. "The H1 and H2 regions of the activation domain of herpes simplex virion protein 16 stimulate transcription through distinct molecular mechanisms, Ikeda et al., Genes to Cells (2002). Prediction: If p300/CBP recruitment is essential for positive feedback, then you will no longer see a bistable response. If you see bistability, then an alternative should be considered more seriously.

We really appreciate the constructive suggestions on this and the following specific experiments. We performed single-cell mRuby dynamics experiments. The results (Figure 2F-G) validate the biostability model with relative stable “high” and “low” mRuby expression states and low probabilities of stochastic transition between the states. We chose the cells within a narrow range (all cells within 2-folds of GFP-GAVPO) and indistinguishable between the “high” and “low” cells (Figure 2H). We further validate the hysteresis by adding another 5xUAS-mCardinal-PEST reporter to HeLa-AB1 cells to increase its temporal sensitivity with a destabilized mCardinal. The cells were either treat with 100 μW/cm^2^ for 8 hours to tilt the 5xUAS local histone acetylation toward "open" states before the 48-hour AM light induction experiment. The control experiment would be kept the cells in the dark before the 48-hour experiments. As predicted with hysteresis, both the mRuby and mCardinal distributions for intermediate light intensities were tilted toward the "open" peak with the prior 100 μW/cm^2^ AM light treatment (Figure 2I-J).

The GAVPO only contains the activation domains of the p65 (a.k.a. relA). There are two papers that provide direct biophysical measurements of the effects of point mutations on p65AD on interactions with CBP. We generate two point-mutations, L449A/F473A, in the TA2 section on GAVPO according to https://journals.plos.org/plosbiology/article?id=10.1371/journal.pbio.1001647. Bimodal distribution disappeared with a new HeLa cell line constructed with this GAVPO mutant (Figure 3G-H), as predicted with GAVPO-CBP/p300 positive feedback loop. Additional mutation F542A on the TA1 section also disrupts its binding to other critical transcriptional factor complexes such as TFIIH, according to https://academic.oup.com/nar/article/45/9/5564/3057346. This might explain the inability to isolate HeLa cell clones with triple mutation L449A/F473A/F542A in GAVPO.

2. Use a monomeric transcription factor regulated by a chemical (e.g. Zinc-finger-ER-p65) and engineer a mRuby promoter with a single binding site. Prediction: This system should exhibit bistable mRuby response at intermediate hormone concentrations, if your model is correct. NOTE: The authors already use an rtTA doxycycline-inducible system in Figure 2—figure supplement 2, which has VP16 activation domain, and exhibits a bimodal distribution at intermediate doxycycline concentrations. The authors could modify VP16 to abolish p300/CBP recruitment of the rtTA construct.

To compare with vast data in the paper, instead of test mutation with VP16, we decided to replace the VP16 in the rtTA transactivator of the tetON system with p65AD. It also exhibits distribution with increasing doxycycline concentrations (Figure 1—figure supplement 4C), similar to LightOn cells with increase light inductions. When we reduce the tetO number from seven to one, we really appreciate the constructive suggestions on this and following specific experiments. The folds of induction (dynamic range) were reduced from 200 to 6, too low to define the existence of bistability clearly. Further studies with new developments in single-cell ChIP-seq with better coverage could help directly assess this question.

2) In its current form, Figure 5 does not convincingly show that PWM reduces noise:(i) In Panel E, at time 0 (no light induction), AM 100 uW, 25 uW and PWM start at different CV values. It is unclear why this is the case especially since this is not observed in the mRuby CV. This needs to be addressed.(ii) The authors should add a panel to Figure 5 showing the mean mRNA levels for the three respective cases, AM 100 uW, 25 uW and PWM 200m.(iii) A discussion should be included about the large number of cells showing almost no mRNA for AM 25 uW and PWM 200m, does this means that there are cells trapped in the off state, and how do these cells impact the CV calculations?(iv) Since the pulsatile behavior appears in ~15% of the cells (Figure 5C), how does the CV time course behave if CV is only calculated for these cells exhibiting pulsatile behavior.

We appreciate the suggestion. We realize that “the “off” and “on” epigenetic states” should be changed to “the “low” and “high” epigenetic states” in Line 127 of the original manuscripts, in consistent with the experimental measurement of H3K27ac (Figure 3J).

(i) The setting up of cells for the multi-positional live-cell imaging on spinning disk confocal requires multiple moving, adjustment, and checking. The room hosted another confocal microscopy that is also heavily used. Although we paid extra attention, it is tough to ensure the cells were absolute not exposed to ambient light before the experiment. A small fraction of cells might have sporadic nuclear mRNA detected at the beginning, which would influence CV calculation when most of the cells don't have nuclear mRNA. We repeated two experiments with AM 25 mW since the original ones were possibly more influenced by ambient light before the experiment.

(ii) We added a panel with mean mRNA levels of the three respective cases, as in Figure 5E.

(iii) The original linear scale plot in Figure 5B-D didn't represent well the true mRNA distribution. The percentage of cells don't have detected nuclear mRNA during the 1000 min are 8%, 20%, and 0.6% for AM 100 mW, 25 mW, and PWM 200 min, respectively. We change the plots to the nature logarithm scale to better represent the distribution of detected mRNA. The number of MCP-tdTomato bound to each mRNA is most likely not identical. Take into consideration of other detection variations. There will be a fraction of nuclear mRNA that failed to be detected in our experiments. We don't have access to better methods at this moment, and the leading labs that developed this technology also use similar spinning disk confocal microscopy. Lightsheet microscopy could increase the signal-noise ratio of these images and improve the detection efficiency. Unfortunately, we don't have access to such microscopy that can track a large number of cells, which is necessary because most of the cells would be out of bound at one time or another due to cell motility. We recently learned a newly developed smFISH method that could be adapted for live cells, but the details are not available at this time. Much more cells need to be analyzed to identify the threshold of “low” and “high” states with nuclear mRNA counts alone. The challenges are detection efficiency and cell motility. The single nuclear mRNA assay RNA The CV calculation is not sensitive to cells with zero mRNA unless the extreme scenario that almost all cells have no mRNA and a few cells with many mRNAs, which would skew the CV value. Our data weren’t the case.

(iv) With the data presented in logarithm scale, we can notice that many cells exhibit pulsatile behavior (Figure 5C). This can be further visualized with averaging over subpopulations of cells. As shown in Figure 5G in the revised manuscript., there are approximated 57% of cells show oscillations.

3) Figure 1E/F and around line 120: The argument of 'limited noise of GFP-GAVPO' requires more details, namely a quantitative analysis of this noise and the effects of noise propagation – the current analysis focuses on relations between means and it is not clear what the effect of GAVPO expression noise on mRuby noise is. In addition, TetR-GFP-nuc may not accurately reflect GAVPO noise if the stabilities of the two proteins are different.

A revised analysis with quantitative analysis is presented in Line 125-137 of the revised manuscript. In Benzinger and Khammash’s model, the mapping from transcriptional factor concentration to gene expression follows a sigmoidal function, but not experimentally presented (Figure 3D in Benzinger and Khammash, 2018). We calculated the mapping curve of GFP-mRuby in AM with 25 mW/cm^2^ light, defined by their model (red line in the center panel of Figure 1H). We agree that the TetR-GFP might not accurately reflect GAVPO noise. We would like to have GFP-GAVPO fusion protein. However, the GAVPO with the current setting would need to be maximum expressed to ensure sufficient mRuby expression at intermediate light. The interference of GAVPO function with fusion protein would be critical for the experiments. If only the static rates of degradation are different and assuming linear degradation process, the TetR-GFP and GAVPO would be well-correlated. The mRNA (small number, influenced by epigenetics, transcriptional bursting, etc.) is more critical for gene expression heterogeneity than protein with an abundant number.

4) TetR-noise reducing circuit (or Plasmid A1). The design or purpose of this plasmid is a little confusing. The TetR-GFP-nuc synthetic transcription factor binds to its CMV promoter to repress its own transcription (negative feedback loop). This should repress mean gene expression and "squeeze" the expression variance about the mean (ref: Becskei et al., Nature 2000), i.e. noise reducing. However, the authors added doxycycline one day before the start of light induction and then maintained it throughout the experiment. This would destroy the negative feedback loop and, thus, increase the TetR and GAVPO mean and variance back to unrepressed CMV promoter levels. Are the authors using this circuit to keep TetR and GAVPO low until the start of the experiment (conditional expression) or are the authors using this circuit to reduce variance in TetR and GAVPO levels (noise-reduction)? Please clarify how and why you are using this circuit.

The TetR-noise reduction circuit was initially chosen to tuning GAVPO expression while maintaining a low noise. During the construction of cell lines, we found that comparing to CAG-GAVPO (plasmid A2), this circuit struggled to provide sufficient GAVPO unless it is maximumly induced, probably due to the addition of TetR-GFP-T2A to the 5' side reduced GAVPO expression level. Therefore, in all experiments, we use 1 mg/mL dox. A sentence was added to lines 110-111 in the revised manuscript.

5) Although not essential in a revised manuscript, testing systems beyond light-regulated GAVPO would strengthen the generality of the authors' conclusions. The good news is that their rtTA system (which has a VP16 trans-activation domain that recruits both p300/CBP and RNA Poll II) exhibits bimodal distribution at intermediate concentrations of doxycycline (Figure 2—figure supplement 2), so they have a different system, strains, and preliminary results to test the generality of their model.

We appreciate your suggestion. Unfortunately, we had performed all the additional experiments suggested earlier. Most of them involve the construction of new plasmids and generate new cell lines and analysis, and with limited human resources at disposal. We didn’t have a chance to test this idea and would love to do it in the next project.

Presentation:6) The text requires thorough language editing throughout. Widespread grammatical mistakes-in particular on verb tense and singular/plural-make it difficult to follow the arguments.

Sorry to cause the trouble. We neglected one round of grammatical checks with the last submission. We hope that we corrected this mistake this time.

7) If the authors decide to include a calibrated mathematical model, the model should be more clearly described in the main text (not only in the Supplementary Material).

We tried stochastic simulation to calibrate our model with experimental data. It proved to be too time-consuming to fit the experimental data. We decide to accept the limitation and deemphasize the current model.

8) A brief description of how the light inducible GAVPO system works would be helpful either in the introduction or the beginning of the Results section. Perhaps adding Figure 1—figure supplement 1 to Figure 1.

Added as Figure 1A in the new manuscript

9) Please add a schematic in Figure 1 or the supplement that clearly illustrates how the light is pulsed (i.e., duration of on time and off time), the light intensity used per pulse, and the mean light intensity. Something similar to Figure 4 D-E, yet for all period lengths.

We added Figure 1B in the new manuscript as suggested. Thanks.

10) Please explain and/or motivate the use of different PWM regimes to provide evidence for different aspects of the work, e.g., line 137 400min period, line 261 600min, line 296, 200min).

We performed PWM with a period of up to 800 min and observed noise reduction and used the maximum of 400 min for most of the flow cytometry assays. The choice of 600 min PWM in scATC-seq experiment (line 354 in the revised manuscript) is due to the limited sample number and resources available while cover approximately one day (1350 min) of the experiment. This choice is a compromise between data points and resource limitation. So it is not explained in the manuscript. The choice for 200 min PWM in mRNA imaging experiment (Line 392 in the revised manuscript) is to help recognize periodicity in mRNA dynamics with more repeats while still exhibit noise reduction at protein level (Figure 1—figure supplement 3B). Due to the photobleaching of tdTomato, we can only take 100 3D imaging stacks for each cell. With a choice of 10 min intervals for the temporal resolution, we can only perform live-cell mRNA imaging for 1000 min. 1000 min only ensures two cycles of 400min PWM, not enough to assure recognition of periodicity. It would cover five cycles of 200 min PWM. The explanation is added to lines 410-412 in the revised manuscript.

11) Figure 2—figure supplement 1 is a confusing figure because it makes two different points. The first half (A-F) shows that LMK-235 has a minor impact on noise reduction. The second half (G-L) shows that a different strain F9-AB2 shows identical signatures to the HeLa-AB1 strain, i.e. bimodality at intermediate levels of AM light, which is reduced to unimodal by FM light. Perhaps split these data into separate supplementary figures? In lines 205 – 207, these data are referred to as "figure supplement 2".

We split it to Figure 2—figure supplement 1 and Figure 2—figure supplement 4, in the revised manuscript as suggested. Thanks.

12) In mouse embryonic stem cells (Figure 1—figure supplement 3), there is only a clear noise reduction effect with 200 and 400 minutes, yet for these there is also a decrease in mean GFP levels. Therefore, the ability to modulate noise independent of mean seems not to hold in mouse embryonic stem cells. Please state this more clearly (text lines 149-154).

A discussion is added at Line 167-171 in the revised manuscript. Thanks.

13) Please provide more detailed explanations of the somewhat puzzling chromatin state dynamics in Figure 4 where a longer-term drift (denoted as epigenetic 'memory') is apparent. How do these dynamics relate to the data presented in Figure 1 (the Methods seem to suggest a measurement time at 48h, but this is not specified in the captions).

There is indeed a long-term drift of chromatin openness in the first 1350 min for both “on” and “off” phase. The protein dynamics data (Figure 1I in the revised manuscript) indicates that the mean expression level reaches a plateau at around 2000 min for PWM. So we postulate that the chromatin openness will also reach plateaus for both “on” and “off” phase.

14) The title 'RNA dynamics flattens PWM-induced pulsatile chromatin opening' implies an influence of RNA dynamics on chromatin state, which is not plausible given the construction of the synthetic system (the mRNA does not encode a protein that could modify chromatin state) and the postulated model (positive feedback is established via TF binding, not mRNA expression). mRNA dynamics will impact protein dynamics, and the title and text should reflect this aspect.

We modified it as suggested, shown in Line 381 in the revised manuscript. Thanks.

15) Figure 2B and Line 943: If there are only 4 data points from 4 independent experiments, why not show them rather than summarize their statistics with a box-whisker plot?

Correct the figure as suggested. Thanks.

16) Line 303/Figure 5C: The export of mRNA from the nucleus to the cytoplasm, which is mentioned in the text, is not shown in the figure that is cited.

We removed ”export of mRNA from the nucleus“ as it is an explanation, not direct observation, located in line 404 in the revised manuscript. Thanks.

17) Line 248: cite Figure 3H in addition to Figure 3—figure supplement 1A.

We corrected in line 340 in the revised manuscript. Thanks.

18) Please write out "min" instead of "m" for the pulse periods in the figure legends.

We changed them as suggested. Thanks.

19) Prior work on the use of PWM and available explanations for its noise-reducing effect should be mentioned and discussed more explicitly.

A new paragraph started at line 517 in the revised manuscript was added. Thanks.

20) A publication that came out recently and also describes independent control of mean expression and variability could be cited: https://doi.org/10.1038/s41467-020-20467-8.

We added it in lines 488-491 in the revised manuscript. Thanks.

[Editors' note: further revisions were suggested prior to acceptance, as described below.]

Thank you for performing additional experiments. The manuscript has been improved but remaining issues make the paper not acceptable for publication in eLife in its current form.In particular, all reviewers felt that bistability of the system is still not well enough supported to make this a major claim of the paper.On the other hand, your findings on PWM-mediated noise suppression in mammalian cells and the link to histone modifications are interesting and large enough an advance for publication. Your data strongly support the idea that CBP/p300 recruitment via chromatin acetylation leads to bimodal gene expression at intermediate GAVPO activity and that PWM of GAVPO activity via light can sculpt the bimodal output to be unimodal.While you could perform additional experiments to attempt to demonstrate bistability (see under point 2 below), we recommend to remove this claim from the paper, replace "bistable" with "bimodal", remove the ODE model, and discuss your findings in a more balanced way that leaves room for alternative interpretations.

Thank you for your positive opinion of our work. We appreciate your efforts to improve our manuscript. We accept the judgments on the bistability. We modified our manuscript according to your general and specific requirements. We removed the ODE model (appendix 2), related Figure 2—figure supplement 5, and lines 284-295 in the previous manuscript, the text described the model in the result sections. The other changes are described in the following paragraphs.

We think the technical difficulty to further validate bistability lies in the long half-life (~1 day) and “response time” of the fluorescent proteins in mammalian cells. In E coli, although the half-life is also relatively long, the fast cell division (~30 minutes) effectively drastically decreases the response time of reporter fluorescent proteins. Although we can use protein tags with a faster degradation time than the “PEST” (estimated to be about 6 hours on our hand), it also drastically reduces the readout and signal-to-noise ratio. We will take advantage of technical development in reporter proteins, and imaging technologies (such as the recently announced super sensitive qCMOS camera), and revisit this issue in the future.

If you want to maintain your conclusion regarding bistability with the current data, you will need to submit to another journal as a home for your paper.Necessary changes for a revision of your paper for eLife include:– Re-writing the second half of the abstract, lines 23 – 29.

We re-wrote this part of the abstract, especially to modify the expressions related to biostability. Please find it in lines 23-30 of the revised manuscript. We also changed the names of institutes to the most updated ones.

– Re-writing the end of the Introduction, lines 83 – 96.

We re-wrote this part of the introduction, especially to modify the expressions related to biostability and positive feedback loop according to the reviewers’ general opinions. Please find it in lines 88-101 of the revised manuscript.

– Considering the criticism of the data in Figure 1H (see below) and modify that section (line 123-137).

We are sorry for not making our analysis clear enough on this part. We appreciate that the reviewer provides his/her matlab code to make sure he/she understood our method. We provided the GFP and mRuby data containing 10640 cells for figure 1G, a new method section: **“**Predictions of mRuby from GFP-GAVPO with single-valued propagation functions”, and the matlab code to generate the predictions in Figure 1H in the new appendix 2. We also performed additional analysis, as new figure 1—figure supplement 5. The matlab code provided by the reviewers, is close to ours, except that our “empirical” and “hypothetical propagation functions” are in linear scale, but plotted in log scale. Noises with lognormal noise were added in this version is different than the reviewer’s version to closely mimic experiments (lines 793-794 in the revised manuscript and appendix 2). With these new analyses, the text describing this (lines 125-137 in the previous manuscript) is revised to lines 129-146 in the revised manuscript.

In this part, we explained that single-valued propagation functions from GAVPO to mRuby expression according to Benzinger and Khammash’s hypothesis cannot recapitulate our experimental observation in two aspects. First of all, although a hypothetical hill function with high cooperativity in GAVPO could mimic bimodal mRuby distribution, our experimental derived propagation function (empirical function) doesn’t show cooperativity at all at the GAVPO concentration (Figure 1H). Secondly, when we split the experimental data into 20 bins according to GFP signals, some of the bins exhibits tight GFP distributions (CV <<0.05) but bimodal mRuby distribution (CV ~1.7). None of the single-valued propagation functions, hypothetical or empirical, can generate such bimodal mRuby distributions (Figure 1—figure supplement 5). It means that a single GAVPO concentration could lead to multiple values of mRuby (plus noise). Putting it together, we exclude the possibility of our experiments follow either Benzinger and Khammash’s hypothesis or any single-valued propagation function.

– Line 227 – 236: Make it clear that this is a working model, but other models are possible. (This may then be better suited for the discussion.)

We re-wrote this part of the result, notes that it is one of the potential models. Please find it in lines 247-262 of the revised manuscript. We discussed other possibilities in discussion (lines 544-546 in the revised manuscript)

– Line 237 – 273: Please see comments under point 2 below. These experiments do not strongly support bistability and the conclusions need to be re-phrased. The model (lines 284-295) could be removed.

We agree with the reviewers that the bistability is not validated sufficiently enough. We rephrased this part to reflect this. (lines 247-262 in the revised manuscript). We also removed part of the ODE model.

– Modify the discussion in lines 493 – 507 and 518-539, since these conclusions are not well supported.

We modified the discussion in lines 493-507 of the previous manuscript to change the phrases on the bistability. Positive feedback loop and hysteresis to reflect that we didn’t sufficiently validate them, and there are other possibilities (lines 515-526 in the revised manuscript). The discussion in lines 531-593 of the previous manuscript is removed for the same considerations.

For the exclusion of Benzinger and Khammash's hypothesis for our experiments, we believe that our new analysis (Figure 1—figure supplement 5) and discussions are sufficient. This discussion in lines 518-531 of the original manuscript is rewritten as lines 547-562 in the revised manuscript.

– Discussion paragraph starting in line 540 needs an introduction.

Thanks for pointing out the omission. We provided an introduction. (lines 572-573 in the revised manuscript).

– More of the limitations in the experiments that you discuss in the response to the reviewers (e.g. point 2, 3, 10, 13 in the previous decision letter) could show up in the paper itself, so that readers are made aware of those.

As suggested, points 2,3(and 10) and 13 are added to lines 437-445, 565-571, and 392-396 in the revised manuscript, respectively.

– Generally, re-assess all mentions of bistability, hysteresis and positive feedback loop.

We modified the description and phrasing of these three concepts throughout the revised manuscript.

Note that, while we point out specific passages to modify, you may want to re-consider the structure of your paper given these major changes.

We split the first section of the results into two sections, re-organized the Discussion section. The text in lines 274-283 and Figure 2—figure supplement 4 was removed because this is a further experiment to test the boundary between bistability and high monostability. Since bistability is not a conclusion of the paper, this part is no longer necessary.

Specific criticisms:(1) Figure 1H aims to demonstrate that bistability and not cooperativity in gene expression (as in Benzinger and Khammash) underlies the observed population distributions by propagating GAVPO distributions through empirical (from Figure 1G data) and hypothetical high-cooperativity transfer functions to predict mRuby distributions. Details on methods are missing (and should be provided), but clearly the mRuby monomodal distribution for the empirical transfer function does not match (approximately) the distribution for AM, 20\muW/cm^2^ in Figure 1F and it is unclear if it does so for the mRuby marginal in Figure 1G (the raw data for this panel is not available), as would be expected. Furthermore, it is unclear, how the predicted mRuby marginal for high cooperativity was obtained. Very approximate simulations for both scenarios (Matlab code below) rather indicate the opposite of the authors' conclusions when compared to data in Figure 1F,G.Matlab code on point (1):%% cooperativity simulations (x: GAVPO, y: mRuby)ns = 1e4;d0 = table2array(readtable('88763_1_data_set_2024261_qtg66v.xlsx','sheet','Fig1H-3','range','a2:b122'));cdf = cumsum(d0(:,2));[~,idx] = unique(cdf);x = interp1(cdf(idx),d0(idx,1),rand(ns,1));% loop: cooperativityhill = {@(x,k,n) 3 + 1*(x-4.5);.…@(x,k,n) 3 + 2*x.^n./(k.^n+x.^n)};n = 70;k = 5.7;figure();for z = 1:length(hill)subplot(3,2,z)xi = linspace(4.5,6.5,100);plot(xi,hill{z}(xi,k,n));subplot(3,2,z+2)y = hill{z}(x,k,n) +.25*randn(size(x));ksdensity([x,y]);view(2);axis([4.5 6.5 2 6])subplot(3,2,z+4);histogram(y,'Normalization','pdf');hold on;histogram(x,'Normalization','pdf');end

We are sorry for not making our analysis clear enough on this part. We really appreciate that you even provide your matlab code to make sure you understood our method. We provided the GFP and mRuby data containing 10640 cells for figure 1G, a new method section: “Predictions of mRuby from GFP-GAVPO with single-valued propagation functions”, and the matlab code to generate the predictions in Figure 1H in the new appendix 2. We also performed additional analysis, as new figure 1—figure supplement 5. The matlab code provided by the reviewers, is close to ours, except that our “empirical” and “hypothetical propagation functions” are in linear scale, but plotted in log scale. Noises with lognormal noise were added in this version is different than the reviewer’s version to closely mimic experiments (lines 793-794 in the revised manuscript and appendix 2). With these new analyses, the text describing this (lines 125-137 in the previous manuscript) is revised to lines 129-146 in the revised manuscript.

In this part, we explained that single-valued propagation functions from GAVPO to mRuby expression according to Benzinger and Khammash’s hypothesis cannot recapitulate our experimental observation in two aspects. First of all, although a hypothetical hill function with high cooperativity in GAVPO could mimic bimodal mRuby distribution, our experimental derived propagation function (empirical function) doesn’t show cooperativity at all at the GAVPO concentration (Figure 1H). Secondly, when we split the experimental data into 20 bins according to GFP signals, some of the bins exhibits tight GFP distributions (CV <<0.05) but bimodal mRuby distribution (CV ~1.7). None of the single-valued propagation functions, hypothetical or empirical, can generate such bimodal mRuby distributions (Figure 1—figure supplement 5). It means that a single GAVPO concentration could lead to multiple values of mRuby (plus noise). Putting it together, we exclude the possibility of our experiments follow either Benzinger and Khammash’s hypothesis or any single-valued propagation function.

(2) Figure 2 aims to demonstrate bistability and hysteresis as proposed by the ODE model via single-cell time lapse microscopy (2F-H) and FACS (2I,J). The single-cell trajectories in Figure 2G, however, do not represent the population distribution. They were selected according to 'low' or 'high' mRuby signal at 24h; the data only demonstrates the dynamics for reaching the target state, and not bistability. For experiments demonstrating hysteresis, in the regime of bistability, one would expect distinct (bimodal under noise) distributions close to the 'low' or 'high' starting state. This is clearly not the case given the data in Figure 2I,J. The slight shift to higher mRuby in the intermediary regime (10-25 light intensity) used as an argument for hysteresis in the manuscript can be easily explained by the population not being in steady-state because the distribution for the predicted monostable 0 light input case is right-shifted by prior stimulation as well.In brief, the single-cell experiments (Figure 2F-H) show a bimodal induction response. The OFF cells stay OFF and the ON cells stay ON. It would have been more convincing to see an induced ON cell stochastically cross a threshold and then return to the OFF state. Or the authors could have started with a fully induced population (100% ON with 100 uW/cm^2^ induction) then put at intermediate light and filmed single cells. The expectation would be that a fraction of the cells crosses the threshold and goes to the stable OFF state. Alternatively, the authors could have sorted ON and OFF cells, and then measured the time evolution of mRuby/mCardinal distribution of the OFF population and ON population.

We accept the judgments on the bistability. We modified our manuscript according to your general and specific requirements. We removed the ODE model (appendix 2), related Figure 2—figure supplement 5, and lines 284-295 in the previous manuscript, the text described the model in the result sections. We made additional corrections based on removing the bistability as a result of the manuscript and made room for alternative possibilities.

We think the technical difficulty to further validate bistability lies in the long half-life (~1 day) and “response time” of the fluorescent proteins in mammalian cells. In E coli, although the half-life is also relatively long, the fast cell division (~30 minutes) effectively drastically decreases the response time of reporter fluorescent proteins. Although we can use protein tags with a faster degradation time than the “PEST” (estimated to be about 6 hours on our hand), it also drastically reduces the readout and signal-to-noise ratio. We will take advantage of technical development in reporter proteins, and imaging technologies (such as the recently announced super sensitive qCMOS camera), and revisit this issue in the future.